# NeurIPS 2024 ML4CFD Competition: Results and Retrospective Analysis

**Mouadh Yagoubi**[1,2]**, David Danan**[1]**, Milad Leyli-Abadi**[1]**, Ahmed Mazari**[3]**,**
**Jean-Patrick Brunet**[1]**, Abbas Kabalan**[4]**, Fabien Casenave**[4]**, Yuxin Ma**[5]**,**
**Giovanni Catalani**[6,7]**, Jean Fesquet**[7]**, Jacob Helwig**[8]**, Xuan Zhang**[8]**, Haiyang Yu**[8]**,**
**Xavier Bertrand**[6]**, Frederic Tost**[6]**, Michael Bauerheim**[7]**, Joseph Morlier**[7]**, Shuiwang Ji**[8]

[1] IRT SystemX, France [2] Technology Innovation Institute, UAE
[3] SimAI team, Ansys Inc, France [4] SafranTech, France
[5] Ant Group, China [6] Airbus, France [7] ISAE Supaero (DAEP), France
[8] Department of Computer Science & Engineering, Texas A&M University

## Abstract

The integration of machine learning (ML) into the physical sciences is reshaping computational paradigms, offering the potential to accelerate demanding simulations such as computational fluid dynamics (CFD). Yet, persistent challenges in accuracy, generalization, and physical consistency hinder the practical deployment of ML models in scientific domains. To address these limitations and systematically benchmark progress, we organized the ML4CFD competition, centered on surrogate modeling for aerodynamic simulations over two-dimensional airfoils. The competition attracted over 240 teams, who were provided with a curated dataset generated via OpenFOAM and evaluated through a multi-criteria framework encompassing predictive accuracy, physical fidelity, computational efficiency, and out-of-distribution generalization. This retrospective analysis reviews the competition outcomes, highlighting several approaches that outperformed baselines under our global evaluation score. Notably, the top entry exceeded the performance of the original OpenFOAM solver on aggregate metrics, illustrating the promise of ML based surrogates to outperform traditional solvers under tailored criteria. However, this does not imply that the winning solution could replace the OpenFOAM solver or that it was overall superior, even for this specific task. Drawing from these results, we analyze the key design principles of top submissions, assess the robustness of our evaluation framework, and offer guidance for future scientific ML challenges.

## 1 Introduction

Machine learning (ML) techniques, and deep learning (DL) in particular, have achieved transformative breakthroughs in domains such as computer vision, natural language processing, and speech recognition. Their influence is now extending to the physical sciences, where ML-based surrogate models offer the potential to accelerate computationally intensive tasks like those in computational fluid dynamics (CFD) (see e.g., [1, 2, 3, 4, 5]). By replacing or augmenting parts of traditional solvers, ML can significantly reduce simulation time, making high-resolution analysis feasible for real-time design and decision-making. However, real-world deployment poses challenges in stability, generalization, and physical consistency. CFD remains a critical bottleneck in many industrial workflows due to its high computational cost. Each simulation can take hours to run, particularly when resolving turbulent flows or boundary layers with fine spatial and temporal resolution. This hinders large-scale design space exploration, optimization, and uncertainty quantification. Moreover, CFD fields are

39th Conference on Neural Information Processing Systems (NeurIPS 2025) Track on Datasets and Benchmarks.

inherently multi-scale: while pressure tends to vary more gradually than velocity in many regions, sharp gradients can still arise near leading edges, separation zones, or areas of adverse pressure recovery. In contrast, velocity fields often exhibit steep gradients within boundary layers, making surrogate learning particularly difficult in these regions. Bridging the gap between ML research and industrial deployment remains a key challenge. While the fusion of ML and physics is a vibrant area, much of the work remains limited to academic benchmarks or idealized settings. Initiatives like the *Machine Learning and the Physical Sciences* workshop [6], and competitions in molecular simulations [7], particle physics [8], and robotics [9], reflect growing interest in translational ML. Recent efforts (e.g., [10, 11, 12, 13]) show how incorporating physical priors can improve robustness and interpretability. Still, few benchmarks address deployment bottlenecks in high-fidelity industrial CFD, where accuracy, latency, and generalization must be balanced. To address this, we organized a competition focused on a representative CFD task with industrial relevance: steady-state aerodynamic simulations of 2D airfoils, using a dataset designed with real-world constraints in mind. Participants predicted velocity, pressure, and turbulent viscosity from mesh-based inputs while ensuring physical consistency. Evaluation used the LIPS (Learning Industrial Physical Simulation) platform [14], combining ML metrics (accuracy, inference time) and physics-aware criteria (e.g., OOD generalization, physical consistency). The competition drew over 240 teams and highlighted a diverse range of methods from PCA-based Gaussian Processes to graph neural networks and coordinate-based neural fields. This methodological variety reflects the difficulty of building surrogates that are simultaneously accurate, fast, and deployment-ready. Generalization across different airfoil geometries, especially under out-of-distribution conditions, emerged as a core challenge. Successful methods often incorporated geometric inductive biases, which proved essential for learning physically coherent surrogates suitable for downstream tasks. In this retrospective, we revisit the competition's structure, results, and lessons learned. We analyze the diversity of submitted approaches, assess the evaluation framework, and discuss practical trade-offs for ML deployment in CFD. Our goal is to guide the development of surrogates that are not only high-performing, but viable for real-world applications.

## 2    Competition Overview

**The airfoil use-case**   We consider a canonical benchmark in computational aerodynamics: the simulation of steady, incompressible, subsonic flow past 2D airfoils under high Reynolds number conditions. While originally framed as a shape optimization task seeking geometries with optimal lift-to-drag ($C_L/C_D$) ratios, our objective shifts towards data-driven emulation of the underlying physics. Specifically, we aim to learn surrogates that approximate the solution operator mapping geometric and flow boundary conditions to the full-field solution of the incompressible Reynolds-Averaged Navier–Stokes (RANS) equations. All geometries are parameterized by the NACA 4- and 5-digit series [15], operating at sea-level, $T = 298.15$ K, with Reynolds numbers ranging from $2 \times 10^6$ to $6 \times 10^6$, ensuring subsonic Mach numbers ($M < 0.3$) and preserving incompressibility assumptions [16]. Despite the 2D simplification, the regime remains highly nontrivial due to transitional and turbulent boundary-layer dynamics, particularly near-wall anisotropy and multi-scale interactions that emerge at high Reynolds numbers. Capturing these effects is essential for accurate prediction of full flow fields (velocity, pressure, turbulent viscosity) and integrated quantities such as lift and drag coefficients. These interactions pose challenges to both conventional solvers and neural surrogates, particularly in extrapolative regimes [17, 18]. Ground-truth simulations are generated using stabilized finite volume methods in OpenFOAM, relying on a structured C-mesh with 250–300k cells and $y^+ \approx 1$ near the wall [16]. The governing PDE system incorporates a two-equation turbulence closure via the SST $k$–$\omega$ model. The ensemble-averaged incompressible RANS equations are:

$$\begin{cases} \partial_i \overline{u}_i = 0 \\ \partial_j(\overline{u}_i \overline{u}_j) = -\partial_i(\frac{\overline{p}}{\rho}) + (\nu + \nu_t)\partial_{jj}^2 \overline{u}_i, \quad i \in \{1, 2\}, \end{cases} \tag{1}$$

where $\overline{\cdot}$ denotes an ensemble averaged quantity, $\partial_i$ the partial derivative with respect to the i-th spatial components, $u$ the fluid velocity, $p$ an effective pressure, $\rho$ the fluid specific mass, $\nu$ the fluid kinematic viscosity, $\nu_t$ the fluid kinematic turbulent viscosity and where we used the Einstein summation convention over repeated indices. A notation table is provided in Appendix A and for more details on the physical setting, we refer to Appendix B.

**Dataset specification and structure**   The dataset is derived from the AirfRANS suite [16, 19], encompassing simulations over a range of aerodynamic configurations with controlled variation in

Reynolds number and angle of attack. Three subsets are constructed to evaluate generalization under different ML regimes:

- *Training set (scarce regime)*: 103 samples drawn from the 'scarce' scenario, filtered to retain only instances with $3 \times 10^6 \le \text{Re} \le 5 \times 10^6$.

- *In-distribution test set*: 200 samples from the 'full' test split, within the same Reynolds regime.

- *Out-of-distribution (OOD) test set*: 496 samples with extrapolated Reynolds ranges ($[2, 3] \cup [5, 6] \times 10^6$), absent from training.

Each simulation is provided as an unstructured point cloud, cropped to $[-2, 4] \times [-1.5, 1.5]$ m, with per-node input features: 2D inlet velocity (m/s), distance to the airfoil (m), and unit normals (m) zeroed outside the surface. Supervision targets include: velocity components ($\overline{u}_x$, $\overline{u}_y$), reduced pressure ($\overline{p}/\rho$), and turbulent viscosity ($\nu_t$), along with a binary mask indicating surface nodes for force reconstruction.

**Challenges in learning and generalization**   From an ML standpoint, the interpolation regime already poses challenges due to the presence of fine-scale gradients, anisotropic boundary conditions, and nonlinear coupling between fields. However, extrapolation particularly in Reynolds number and angle of attack introduces sharp regime shifts that require true inductive generalization, not mere interpolation. Models are expected to remain predictive under geometric and parametric shifts far from the training distribution, making this task a strong testbed for OOD learning [19, 18]. Moreover, generalization *out of design space* (OODS) remains underexplored: even within the NACA families, combinations of camber and thickness parameters may yield aerodynamic regimes unseen during training. This raises the need for inductive biases or learned operators that respect underlying PDE structures [20] and are robust to topological or geometric perturbations.

**Boundary layer resolution and force recovery**   A key difficulty lies in the accurate prediction of boundary-layer profiles near the airfoil surface. These regions exhibit extreme stiffness in the velocity and turbulence fields, with gradients that collapse into micrometer-scale layers. Most ML architectures struggle to preserve these sharp features due to smoothing inductive biases and subsampling strategies. Yet, recovering physically consistent surface stresses (e.g., for drag estimation) critically depends on these predictions [16]. This makes high-resolution field prediction not just integral quantities a core benchmark objective.

**Toward aerodynamic shape optimization**   While the benchmark is framed as a surrogate modeling task, the ultimate goal is gradient-informed shape optimization a highly sensitive inverse problem requiring accurate gradients of drag/lift with respect to geometry. Surrogates that preserve rank correlations of $C_L$ and $C_D$ are useful for coarse search, but optimal control and differentiable design require surrogates to be both physically faithful and geometrically responsive [17]. In this context, inductive architectures like mesh-based GNNs [20, 21] or neural operators [18] are promising candidates due to their ability to encode spatial locality and generalize across domains.

**Competition testbed framework**   For the evaluation process of the submitted solutions, we used the Learning Industrial Physical Simulation (LIPS) benchmarking framework [14]. Primarily, the LIPS framework is utilized to establish generic and comprehensive evaluation criteria for ranking submitted solutions. This evaluation process encompasses multiple aspects to address industrial requirements and expectations. In this competition, we considered the following evaluation criteria:

- *ML-related performance*: we focus on assessing the trade-offs between typical model accuracy metrics like Mean Squared Error (MSE) and computation time;

- *Out-of-Distribution (OOD) generalization*: given the necessity in industrial physical simulation to extrapolate over minor variations in problem geometry or physical parameters, we incorporate OOD geometry evaluation, such as unseen airfoil mesh variations;

- *Physics compliance*: ensuring adherence to physical laws is crucial when simulation results influence real-world decisions. While the machine learning metrics are relatively standard, the physical metrics are closely related to the underlying use case and physical problem. There are two physical quantities considered in this challenge namely, the *drag* and *lift* coefficients, for which we compute two coefficients between the observations and predictions: (1) The spearman

correlation, a nonparametric measure of the monotonicity of the relationship between two datasets (Spearman-correlation-drag: $\rho_D$ and Spearman-correlation-lift: $\rho_L$); (2) The mean relative error (Mean-relative-drag: $C_D$ and Mean-relative-lift: $C_L$).

**Scoring** We propose an *homogeneous evaluation* of submitted solutions to learn the airfoil design task using the LIPS platform. The evaluation is performed using 3 previously mentioned categories: ML-Related, Physics compliance and out-of-distribution generalization. For each category, specific criteria related to the airfoils design task are defined. The global score is computed based on linear combination of the sub-scores related to three evaluation criteria categories, each comprising a set of metrics:

$$Global\_Score = \alpha_{ML} \times \textbf{Score}_{\textbf{ML}} + \alpha_{ood} \times \textbf{Score}_{\textbf{ood}} + \alpha_{Physics} \times \textbf{Score}_{\textbf{Physics}}, \quad (2)$$

where $\alpha$s designate the coefficients to calibrate the relative importance of each categories. For more details concerning the computation of each subscore, the reader could refer to Appendix C.

**Competition resources and execution pipeline** To evaluate participant submissions, we used six high-memory GPUs (two NVIDIA A6000 and four A40, each with 48 GB of VRAM) provided by NVIDIA and IRT SystemX. These were distributed across two identical servers, each equipped with dual Intel Xeon 5315Y CPUs (8 cores, 16 threads, 3.2GHz) and 256GB of DDR4 RAM. Each GPU ran a dedicated compute worker, containerized with access to 10 CPU threads, ensuring isolation and reproducibility. The submission and evaluation pipeline was integrated into the Codabench platform, enabling standardized, fair assessment and consistent hardware conditions, regardless of participants' local computing capabilities. This infrastructure was used to both train and evaluate all submitted models during the competition. To facilitate onboarding, we provided a starter kit with self-contained Jupyter notebooks and sample submissions (available at: `https://github.com/IRT-SystemX/NeurIPS2024-ML4CFD-competition-Starting-Kit`). A dedicated discussion channel supported participants and facilitated communication throughout the event.

## 3 Competition results

A total of 650 submissions were made on Codabench by approximately 240 registered participants or teams. Table 1 presents the competition leaderboard, showcasing the top four awarded solutions. Additional details regarding the competition design are provided in Appendix D. Notably, the method ranked first (MMGP) was submitted in two distinct variants, with the better-performing one outperforming the physical baseline in overall performance. Rankings were determined based on a global score that aggregated performance across several key criteria, including machine learning (accuracy and speed-up), physics, and out-of-distribution (OOD) generalization. Participants proposed a diverse range of approaches, including hybrid ML-physics models, purely machine learning–based techniques such as graph neural networks (GNNs), and mathematical engineering techniques used as a preprocessing step.

Table 1: NeurIPS 2024 ML4CFD Competition Results. The performances reported using three colors computed with respect to two thresholds. Colors meaning: 🔴 Unacceptable (0 point) 🟠 Acceptable (1 point) 🟢 Great (2 points). Reported results: OpenFOAM (Ground Truth), Baseline solution (Fully Connected NN), and the 4 winning solutions. MMGP solution includes 2 variants.

| | | Criteria category | | | | | | |
|---|---|---|---|---|---|---|---|---|
| | | **ML-related (40%)** | | **Physics (30%)** | **OOD generalization (30%)** | | | **Global Score (%)** |
| | Method | Accuracy(75%) | Speed-up(25%) | Physical Criteria | OOD Accuracy(42%) | OOD Physics(33%) | Speed-up(25%) | |
| | | $\overline{u}_x\ \overline{u}_y\ \overline{p}\ \overline{v}_t\ \overline{p}_s$ | | $C_D\ C_L\ \rho_D\ \rho_L$ | $\overline{u}_x\ \overline{u}_y\ \overline{p}\ \overline{v}_t\ \overline{p}_s$ | $C_D\ C_L\ \rho_D\ \rho_L$ | | |
| | OpenFOAM | 🟢🟢🟢🟢🟢 | 1 | 🟢🟢🟢🟢 | 🟢🟢🟢🟢🟢 | 🟢🟢🟢🟢 | 1 | **82.5** |
| | Baseline(FC) | 🔴🔴🔴🔴🔴 | 750 | 🔴🔴🔴🔴 | 🔴🔴🔴🔴🔴 | 🔴🔴🔴🔴 | 750 | **11.09** |
| Rank | | Top solutions | | | | | | |
| 1 | #1 MMGP | 🟢🟢🟢🟢🟢 | 162.7 | 🟢🟢🟠🟢 | 🟢🟢🟠🟢🟠 | 🟢🟢🟠🟢 | 162.8 | **84.68** |
| - | #1 MMGP-2 | 🟢🟢🟢🟢🟢 | 68.9 | 🟢🟢🟠🟢 | 🟢🟢🟠🟢🟢 | 🟢🟢🟠🟢 | 68.9 | **80.54** |
| 2 | #2 OB-GNN | 🟢🟠🟢🟠🟠 | 318.9 | 🟢🟢🟠🟢 | 🟢🟢🟢🔴🟠 | 🟢🟢🟠🟢 | 319 | **77.45** |
| 3 | #3 MARIO | 🟢🟢🟢🟠🟠 | 617.2 | 🟢🟢🔴🟢 | 🟢🟢🔴🔴🟢 | 🟠🟢🔴🟢 | 618.2 | **71.20** |
| 4 | SP : GMP-NN | 🟢🟠🔴🟠🔴 | 513.3 | 🟠🟠🔴🟢 | 🟠🟢🔴🔴🟠 | 🟠🟠🔴🟢 | 513.3 | **50.60** |

The four winning methods, described in Section 4, adopted distinct strategies but shared the common objective of building accurate and efficient surrogate models for CFD, particularly under geometric

variability. All methods included a geometry processing step, implemented either explicitly (e.g., mesh morphing in MMGP or latent graph construction in GeoMPNN) or implicitly (e.g., learned embeddings in MARIO and OB-GNN). MMGP stood out by using a non-deep learning pipeline based on Principal Component Analysis (PCA) and Gaussian Processes, demonstrating that classical ML techniques can still be competitive for surrogate modeling. OB-GNN introduced custom graph convolutions sensitive to spatial offsets, enhancing the model's ability to capture local geometric features. MARIO relied on coordinate-based neural fields, incorporating Fourier embeddings and conditional modulation to disentangle geometric and physical information, while GeoMPNN used a message-passing mechanism to transfer geometric information from the airfoil surface to the volume mesh. These methods varied not only in modeling paradigms (ranging from Gaussian Processes and GNNs to neural fields) but also in interpretability and preprocessing complexity. Despite their differences, all emphasized physical consistency, geometric awareness, and strong generalization across a wide range of airfoil geometries. In terms of speed-up, while MMGP achieved the highest overall ranking, the deep learning methods demonstrated significantly greater acceleration over the physical solver—typically 2–3× faster than MMGP. However, this advantage was not fully captured in the final ranking due to the scoring function's emphasis on accuracy (75%) over speed-up (25%). This suggests opportunities for improving MMGP's runtime performance and highlights the promise of deep learning approaches, which already excel in speed but must continue to close the gap in accuracy. Beyond technical performance, we also analyzed the organizational outcomes of the competition alongside the impact of incentive measures on competition dynamics. More details can be found in Appendices D.2 and D.4.

## 4 Description of Winning solutions

### 4.1 MMGP: a Mesh Morphing Gaussian Process-based machine learning method for regression of physical problems under non-parameterized geometrical variability

MMGP, proposed in [22], relies on four ingredients combined together: (i) mesh morphing, (ii) finite element interpolation, (iii) dimensionality reduction, and (iv) Gaussian process regression for learning solutions to PDEs with non-parameterized geometric variations. MMGP inference workflow is illustrated in Figure 1. First, non-parametrized input meshes are converted to learnable objects using the coordinates of the mesh vertices, considered as continuous fields: in the left column of Figure 1 are illustrated the continuous field of the x-component of the coordinates, naturally featuring vertical iso-values. Then, we choose a deterministic morphing process to transform the input meshes onto meshes of a chosen common shape (the unit disk in the example of Figure 1). At this stage, each sample is converted in a different mesh of the unit disk, and we notice that the coordinate fields have been transformed in a unique fashion for each sample, that depends on the shape in the input meshes. Then, we choose a mesh of the common shape, and we project the coordinate fields of each morphed mesh onto this common mesh, using finite element interpolation. Now, the coordinate fields of all the samples are represented on the same mesh, hence has the same size. We can apply classical dimensionality reduction techniques, like the PCA, to obtain low-dimensional vectors, representations of our mesh. If the learning problem features scalar inputs, we concatenate them at the end of these vectors. Variable size output fields are converted to learnable objects in the same fashion, leading to low-dimensional embeddings.

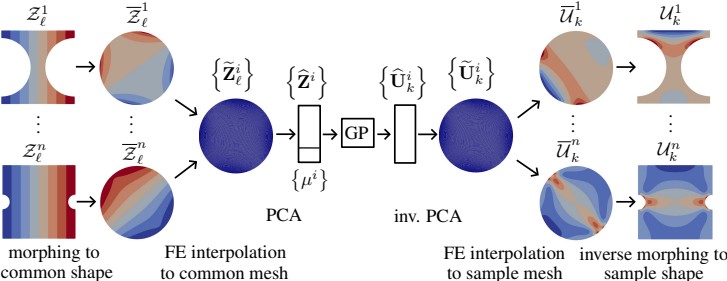

Figure 1: Illustration of MMGP inference workflow for the prediction of an output field [22].

Our entry to the competition is detailed in Section F, and the code is available at `https://gitlab.com/drti/airfrans_competitions/-/tree/main/ML4CFD_NeurIPS2024`. The performance

of MMGP strongly depends on morphing quality. We train a small metamodel to infer the angle of the wake from the input airfoil profiles, angle of attack and input velocity, so that the wake of the output fields are aligned with the horizontal as close as possible. Other improvements exploit relations between velocity and pressure outputs, and implementation optimizations of the morphing and the finite element interpolation (MMGP-2) to improve the speedup, see Section F.

This competition allowed us to try original ideas, that were not included in the winning solution due to time limitations, but we expect them to increase the speedup further. First, one of the two finite element interpolation can be saved by directly computing the morphing from the reference mesh onto each target mesh. The input to the GP model is now the PCA coefficients of the morphing. After predicting the solution on the reference mesh, we morph the reference mesh onto the target mesh and then project the solution using finite element interpolation. Second, instead of resorting RBF morphing in the inference stage, we can construct a reduced basis of the RBF morphings computed on the training set. This leads to compute and factorize a $N \times r$ matrix in the training phase, instead of the $N \times N$ linear systems resolutions during inference. Authors have proposed physics-based models that are compatible with MMGP. The difference is that instead of relying on a data-driven low-dimensional models (the Gaussian processes in the case of MMGP), it is possible to efficiently assemble and solve the Navier-Stokes equations on the low-dimension space spanned by the PCA modes obtained after morphing, leading to a hyper-reduced least-square Petrov-Galerkin scheme [23]. We expect such methods to greatly improve the accuracy.

## 4.2 Offset-based Graph Convolution for Mesh Graph

Inspired by PointConv's coordinate-sensitive convolution design [24], we propose an enhanced graph neural network (GNN) method for CFD simulations. Conventional GNNs struggle with irregular point cloud distributions and spatial coordinate sensitivity, which are critical for physical simulations. Our offset-based graph convolution addresses these challenges through two key innovations: offset-driven kernel weight generation and geometry-aware aggregation. The workflow is illustrated in Figure 2, which shows the convolution workflow and our GNN models.

The input graph structure is constructed by augmenting the original mesh with k-Nearest Neighbors (kNN, k=10) connectivity to mitigate abrupt connectivity changes from artificial mesh segmentation. For each node $i$ and neighbor $j$, we calculate the geometric offset vector $o_{ij} = x_i - x_j$, where $x_i$ and $x_j$ are the 2D spatial coordinates of nodes $i$ and $j$, respectively. This offset vector captures the spatial relationship between the nodes.

A 2-layer MLP transforms this offset into dynamic scaling weights $\theta_{ij} = \text{MLP}_\theta(o_{ij})$. Simultaneously, inverse-distance normalized attention weights are computed as $\alpha_{ij} = \frac{\exp(-\|o_{ij}\|_2)}{\sum_{k \in \mathcal{N}(i)} \exp(-\|o_{ik}\|_2)}$. These weights ensure that closer neighbors have a higher influence on the feature update. The node features are updated using self-update and neighborhood aggregation. The self-update is performed via $\text{MLP}_{\text{self}}(h_i^{(k)})$, while the neighborhood aggregation involves $\sum_{j \in \mathcal{N}(i)} \alpha_{ij} \cdot (\theta_{ij} \otimes h_j^{(k)})$, where $\otimes$

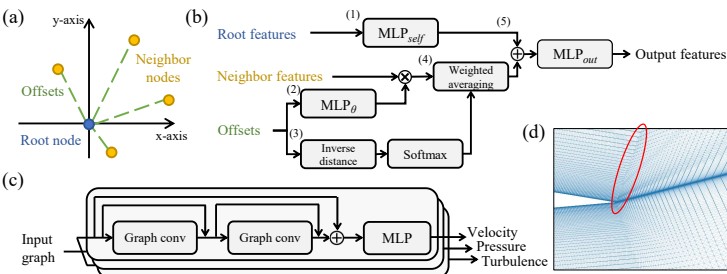

Figure 2: Offset-based graph convolution architecture. (a) Spatial relationships: root node (blue), neighbor nodes (orange), offset vectors (green dashed lines). (b) Convolution workflow: key steps including self-feature update, kernel weight generation, attention weights, aggregation, and feature fusion. (c) Our method uses three dedicated GNN models for different predictions. (d) Hybrid graph construction with kNN connections and abrupt connectivity transition.

denotes element-wise multiplication. The final feature update is achieved through concatenation and a final MLP: $h_i^{(k+1)} = \text{MLP}_{\text{out}}(\text{MLP}_{\text{self}}(h_i^{(k)}) \oplus \sum_{j \in \mathcal{N}(i)} \alpha_{ij} \cdot (\theta_{ij} \otimes h_j^{(k)}))$.

Our method employs three dedicated GNN models for velocity (two axes), pressure, and turbulent viscosity predictions to handle feature conflicts in multitask learning. The velocity/pressure model uses a larger hidden dimension (256) compared to the turbulent viscosity model (128). Each model stacks two offset-based graph convolution layers with skip connections, followed by a multi-layer perceptron (3-5 layers) for target-specific decoding. The training workflow combines boundary-aware neighbor sampling (surface nodes sampled 8× more frequently than other nodes and near-wall nodes 2×), additive Gaussian noise injection, and a hierarchical sampling strategy to preserve local flow structures. A tailored loss function dynamically balances prediction errors and incorporates surface-specific regularization terms. The complete system achieves a 7,000× speedup (220 ms vs 25 min per case) compared to OpenFOAM simulations while maintaining essential physical consistency. This demonstrates that graph-based learning can effectively bridge data-driven efficiency and physics fidelity. The code is available at: `https://github.com/SolarisAdams/ML4CFD-Offset-based-Graph-Convolution`

### 4.3 MARIO: Multiscale Aerodynamic Resolution Invariant Operator

MARIO (Modulated Aerodynamic Resolution Invariant Operator), introduced in [25], is a coordinate-based neural field architecture designed for efficient surrogate modeling of aerodynamic simulations. The framework leverages the resolution-invariant properties of neural fields to accurately predict flow variables around airfoils while handling non-parametric geometric variability. MARIO's modeling strength lies in its conditional formulation that effectively separates geometric information from spatial coordinates through an efficient modulation mechanism. For the competition, geometric information was encoded in a conditioning vector $z_{geom}$ representing airfoil characteristics through thickness and camber distributions at fixed chord locations. This approach was selected for its simplicity and inference efficiency, though MARIO also supports more general meta-learned representations through implicit distance field encoding as demonstrated in previous work on 2D and 3D aerodynamics [26, 27, 28]. Inflow conditions ($V_x$, $V_y$) are then concatenated with geometric conditioning and passed through a hyper-network $h_\psi$ parameterized by a fully connected MLP that produces a set of modulation vectors $\phi = \{\phi_i\}_{i=1}^L$. The inputs of the main network are the spatial coordinates $(x, y)$, the implicit distance ($d_{sdf}$) and two sets of featured engineered fields: (i) the volumetric normals (two components) and (ii) the boundary layer mask field (one scalar). The effect of these additional inputs, as well as the detailed pre-processing procedure is described in detail in the Appendix H. It is important to observe that the effect of these input features has a strong effect on the model performance also due to the relative scarce availability of data for training (approximately 100 samples). With an increasing amount of data to learn from, the utility of these features might reveal to be less crucial. In order to overcome Spectral Bias [29], the main network $f_\theta$ first processes these inputs through a Multiscale Random Fourier Feature embedding (RFF) [29, 26]: the inputs are

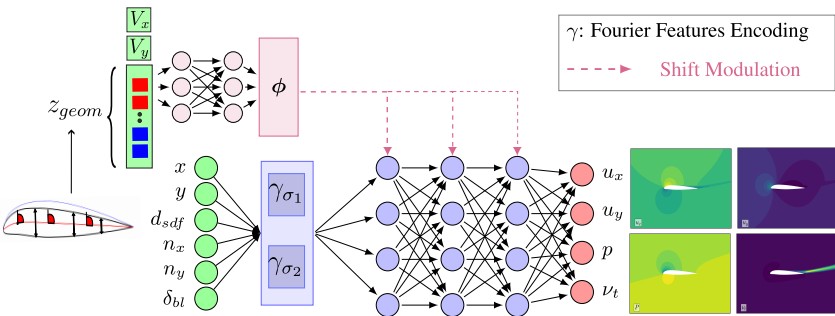

Figure 3: MARIO conditional neural field architecture. Input features (spatial coordinates, SDF, and engineered fields) are processed through multiscale Fourier feature encoding ($\gamma$). The hypernetwork processes the conditioning vector $\mathbf{z}$ (consisting of inflow conditions and geometric parameters) to generate layer-specific modulation vectors $\phi$. These modulations are applied to each intermediate layer of the main network via shift modulation.

mapped using RFFs sampled from multiple Gaussian distrbutions of different standard deviations. The goal is to better capture the frequency content of different output variables. The composition of layers in the network is described as follows:

$$f_\theta(\mathbf{x}) = W_L(\eta_{L-1} \circ \eta_{L-2} \circ \dots \circ \eta_1 \circ \gamma_\sigma(\mathbf{x})) + b_L \qquad (3)$$

$$\eta_l(\cdot) = ReLU(W_l(\cdot) + b_l + \phi_l) \qquad (4)$$

where $f_\theta(\mathbf{x})$ represents the main network parameterized by weights $\theta$ that maps input coordinates $\mathbf{x}$ to output physical fields. The function $\gamma_\sigma(\mathbf{x})$ denotes the multiscale Fourier feature encoding with frequency scales $\sigma$. Each $\eta_l$ represents the $l$-th layer's activation function, with $W_l$ and $b_l$ being the layer weights and biases respectively. The modulation vectors $\phi_l$ (produced by the hypernetwork) are added element-wise before the ReLU activation, implementing the shift-modulation mechanism. An overview of this architecture is provided in Figure 3. During training, we employed dynamic subsampling (using only 10% of the full mesh) to accelerate learning while maintaining the ability to perform full-resolution inference, a key advantage over mesh-dependent approaches. The full implementation of the model is provided in the dedicated Github repository `https://github.com/giovannicatalani/MARIO`. Additional details about the model architecture, hyper-parameters and result on the competition are included in Appendix H.

### 4.4 A Geometry-Aware Message Passing Network for Modeling Aerodynamics over Airfoils

Due to the high influence of the airfoil shape on the resulting dynamics, it is not only key to learn an expressive representation of the airfoil shape, but to efficiently incorporate this representation into the field representations. Furthermore, to handle a large number of simulation mesh points, the design of architectures that are amenable to efficient training strategies is vital. We therefore propose a geometry-aware message passing neural network known as GeoMPNN which efficiently and expressively integrates the airfoil shape with the mesh representation. For efficient training, GeoMPNN is specifically designed to handle subsampled meshes during training without degradation in the solution accuracy on the full mesh at test time. Under this framework, we first obtain a representation of the geometry in the form of a latent graph on the airfoil surface. We subsequently propagate this representation to all mesh points through message passing on a directed, bipartite graph using *surface-to-volume* message passing, visualized in Figure 4.

The latent graph representation of the airfoil geometry is obtained via $L = 4$ layers of learned message passing over the surface mesh $\mathcal{X}_{\text{surf}}$ following a standard message passing scheme [31, 32]. The input surface graph is constructed via a radius graph with radius 0.05 and maximum number of neighbors 8. Input node features $\boldsymbol{z}_{\text{in}}(\boldsymbol{x})$ for mesh point $\boldsymbol{x}$ include the inlet velocity, coordinates, orthogonal distance from the airfoil, and surface normals. Input edge features $\boldsymbol{e}_{\text{in}}(\boldsymbol{y}, \boldsymbol{x})$ for the directed edge from $\boldsymbol{y}$ to $\boldsymbol{x}$ include displacement and distance. The final output of this geometric encoding is a latent representation $\boldsymbol{z}_{\text{surf}}(\boldsymbol{y})$ for all $\boldsymbol{y} \in \mathcal{X}_{\text{surf}}$.

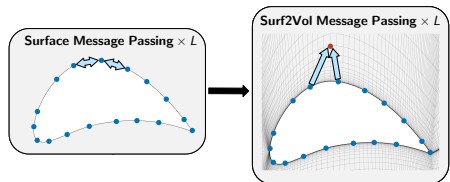

Figure 4: Illustration of surface-to-volume message passing in the GeoMPNN framework [30].

The geometric representation is then propagated throughout the mesh $\mathcal{X}$ using *surface-to-volume* message passing. We first construct a directed bi-partite graph which associates every mesh point with a neighborhood of the latent surface graph using directed SURF2VOL edges from surface points to volume (*i.e.,* the full mesh) points. The SURF2VOL neighborhood $\mathcal{N}_{\text{s2v}}(\boldsymbol{x})$ for mesh point $\boldsymbol{x}$ is constructed by finding the $k = 8$ nearest mesh points on the airfoil surface. For $\boldsymbol{y} \in \mathcal{N}_{\text{s2v}}(\boldsymbol{x})$, node features and SURF2VOL edge features are embedded as $\boldsymbol{z}(\boldsymbol{x}) \coloneqq \text{MLP}(\boldsymbol{z}_{\text{in}}(\boldsymbol{x}))$ and $\boldsymbol{e}_{\text{s2v}}(\boldsymbol{y}, \boldsymbol{x}) \coloneqq \text{MLP}(\boldsymbol{e}_{\text{in}}(\boldsymbol{y}, \boldsymbol{x}))$. In each message-passing layer, edge embeddings are first updated as $\boldsymbol{e}_{\text{s2v}}(\boldsymbol{y}, \boldsymbol{x}) \leftarrow \boldsymbol{e}_{\text{s2v}}(\boldsymbol{y}, \boldsymbol{x}) + \text{MLP}([\boldsymbol{z}_{\text{surf}}(\boldsymbol{y}), \boldsymbol{z}(\boldsymbol{x}), \boldsymbol{e}_{\text{s2v}}(\boldsymbol{y}, \boldsymbol{x})])$, thereby integrating a representation of the airfoil geometry in the form of the surface node embeddings $\boldsymbol{z}_{\text{surf}}(\boldsymbol{y})$ closest to $\boldsymbol{x}$ into the edge embedding $\boldsymbol{e}_{\text{s2v}}(\boldsymbol{y}, \boldsymbol{x})$. This geometric representation is then aggregated into a message $\boldsymbol{m}(\boldsymbol{x})$ via mean aggregation as $\boldsymbol{m}(\boldsymbol{x}) \coloneqq \frac{1}{k} \sum_{\boldsymbol{y} \in \mathcal{N}_{\text{s2v}}(\boldsymbol{x})} \boldsymbol{e}_{\text{s2v}}(\boldsymbol{y}, \boldsymbol{x})$. Because $\boldsymbol{m}(\boldsymbol{x})$ aggregates representations of the airfoil from $\boldsymbol{y} \in \mathcal{X}_{\text{surf}}$ closest to $\boldsymbol{x}$, it can be regarded as an embedding of the region of the airfoil geometry closest to $\boldsymbol{x}$. The local geometric embedding is then integrated into the node embedding of $\boldsymbol{x}$ as $\boldsymbol{z}(\boldsymbol{x}) \leftarrow \boldsymbol{z}(\boldsymbol{x}) + \text{MLP}([\boldsymbol{z}(\boldsymbol{x}), \boldsymbol{m}(\boldsymbol{x})])$. Following the final message passing layer, we apply a decoder MLP to the final latent node representations $\boldsymbol{z}(\boldsymbol{x})$ to obtain the predicted field at mesh point $\boldsymbol{x}$.

To avoid a distribution shift when evaluating GeoMPNN, during training, we do not subsample the nodes of the input surface graph for obtaining the geometric representation $z_{surf}$. As there are less than 1,011 nodes on the airfoil surface on average, the cost of maintaining the full surface graph is minimal. However, since there are more than 179K mesh points per training example, we subsample the volume mesh points to reduce training cost. Because GeoMPNN does not model interactions between volume nodes, there is no shift in the neighborhood structure of mesh points when evaluating the model on the full mesh. We additionally propose a variety of physically-motivated enhancements to our coordinate system representation for the input node and edge features of GeoMPNN, as well as field-specific methods. We provide details on both in Appendix I. The full implementation of GeoMPNN is available at `https://github.com/divelab/AIRS/tree/main/OpenPDE/GeoMPNN`.

## 5 Discussion

**Evaluation protocol soundness and limitations**   Fair evaluation in Scientific Machine Learning (SciML) remains challenging and underexplored. In this competition, we focused on two main goals: (i) ensuring fair and transparent model ranking, and (ii) enabling meaningful comparisons between ML surrogates and traditional solvers. We adopted a linear aggregation scheme combining scores for ML performance, physical consistency, and out-of-distribution (OOD) generalization. During the competition, some participants questioned the definition of the speed-up metric. While a standard formulation considers only inference time, our protocol included both inference and evaluation durations, reflecting that evaluation (e.g., error computation) is not accelerated by ML and should be part of the runtime budget. In industrial workflows, where simulation pipelines include pre- and post-processing steps, such holistic timing is more representative. This choice aligns with Amdahl's law, which emphasizes the bottlenecks from non-parallelizable components. We believe that incorporating evaluation overhead encourages model designs that optimize both predictive accuracy and deployment efficiency. While our protocol offers a rigorous framework, further refinement is possible. As highlighted in [33], benchmarking should also account for training and data generation costs, and comparisons with classical solvers should be framed under either equal runtime or equal accuracy. Standardizing such practices across SciML competitions will be critical for long-term reproducibility and fair benchmarking. More details appear in Appendix E.1.

**Recommendations for designing new competitions in SciML**   Designing SciML competitions requires careful decisions on problem setup and evaluation. A meaningful benchmark must combine a well-posed physical task with relevant evaluation criteria and efficient, validated solvers. Collaborating with domain experts is essential. Offering multiple baselines (classical and ML) and rich metadata fosters reproducibility and avoids misleading comparisons. To mitigate overfitting, organizers should avoid overused datasets, employ multiple evaluation sets, and use hidden test sets for final ranking. The scoring function must reflect task priorities (e.g., speed vs. accuracy), generalize across scenarios, and uphold physical consistency. Post-hoc statistical analysis is valuable to assess ML stochasticity, though often impractical during development. Further guidance appears in Appendix E.2.

## 6 Conclusion

The ML4CFD NeurIPS 2024 competition established a comprehensive benchmark for surrogate modeling in computational fluid dynamics, focusing on steady-state aerodynamic simulations around two-dimensional airfoils. By integrating standard machine learning metrics with physical consistency and out-of-distribution generalization, the evaluation framework enabled rigorous, multi-dimensional comparisons across a wide spectrum of methods. The results revealed that classical machine learning approaches such as MMGP (mesh morphing with Gaussian processes) remain competitive, achieving higher predictive accuracy than many deep learning models. In contrast, neural architectures like OB-GNN, MARIO, and GeoMPNN demonstrated substantial computational speedups ranging from 300× to 600× over traditional CFD solvers highlighting their potential for real-time or iterative workflows. A consistent trait among top-performing methods was the explicit incorporation of geometric structure, whether through mesh morphing, graph-based connectivity, or coordinate-based neural fields, underscoring the critical role of geometric inductive biases in learning physically meaningful surrogates. Nevertheless, the challenge of jointly optimizing accuracy, efficiency, and physical fidelity remains open. Promising research directions include embedding physical constraints directly into training procedures, developing hybrid architectures that combine the interpretability of

classical models with the scalability of neural networks, and improving the capacity of surrogates to capture fine-scale flow features, particularly in boundary layers. Enhancing precision in these regions is crucial for high-fidelity design space exploration and aerodynamic shape optimization, where localized flow behavior often drives global performance outcomes. The competition ultimately highlighted the need for continued innovation at the intersection of machine learning, computational physics, and scientific computing. Moving forward, standardized benchmarks that reflect both computational and physical constraints will be essential for robust and meaningful progress in scientific machine learning.

## Acknowledgments

The authors thank Florent Bonnet for valuable discussions regarding the AirfRANS dataset.

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

# Appendix A   Notation table

Table 2: Notation table

| Notation | Description |
|:---:|:---|
| $L$ | Lift force |
| $D$ | Drag force |
| $C_L$ | Coefficient of Lift force |
| $C_D$ | Coefficient of Drag force |
| $y^+$ | Represents a local Reynolds close to the obstacle |
| $u$ | Fluid velocity (two components along x and y axes) |
| $p$ | Effective pressure |
| $p_s$ | Surface pressure |
| $\rho$ | Fluid specific mass |
| $\nu$ | Fluid kinematic viscosity |
| $\nu_t$ | Fluid kinematic turbulent viscosity |
| $\bar{\cdot}$ | Averaged quantity used for different measures |
| **Scoring formulation** | |
| $\alpha$ | Coefficient used to calibrate the relative importance of different evaluation categories |
| $T_1$ | The first threshold below which the results is considered as great (green) |
| $T_2$ | The second threshold above which the results are considered as not acceptable (red) |
| $\rho_D$ | Spearman correlation of Drag |
| $\rho_L$ | Spearman correlation of Lift |

# Appendix B   Detailed physical setting

**High-fidelity CFD simulation and industrial relevance**   The dataset used in the competition is generated using steady-state, incompressible Reynolds-Averaged Navier–Stokes (RANS) simulations with the $k$–$\omega$ SST turbulence model, solved via the `simpleFOAM` solver in OpenFOAM. The simulations are discretized using the Finite Volume Method (FVM) on structured C-type meshes of up to 300,000 cells. The resolution near the wall ensures $y^+ \approx 1$, allowing the accurate capture of boundary layer behavior. These configurations model realistic subsonic flow over 2D airfoils at Reynolds numbers between $2 \times 10^6$ and $6 \times 10^6$, aligning with industrial aerodynamic regimes. For more details about the physical setting or the physical solver configuration, we refer respectively to [34] and the OpenFOAM setting for this usecase[1]. The latter includes the boundary conditions, thermophysical properties, the detailed finite volume schemes, the linear solvers configuration. While accurate, such high-fidelity CFD simulations are computationally expensive, with each run taking up to 30 minutes. This limits their scalability for use cases like design space exploration, uncertainty quantification, or real-time control. ML surrogates promise significant speedups, but their industrial deployment depends on reproducing both field-level fidelity and physically meaningful derived quantities. Finally, we would like to provide some insights regarding the inherent limitations of the dataset. The airfoil geometries in the dataset are based on the NACA series, which offers a vast, theoretically infinite design space. Small changes in parameters can lead to significantly different aerodynamic behaviors. The same holds true for flow conditions: real-world scenarios are highly diverse and complex, and no single dataset can fully capture that range. As we do have the generator, we could obtain a much larger dataset. However, generating a truly exhaustive dataset would not only be computationally prohibitive but also unlikely to guarantee meaningful gains in model generalization or speed-up under fair evaluation (see Equation 12). Moreover, such a dataset would still fall short of representing every possible real-world scenario. Instead, one of the core challenges of this competition was to reflect a realistic industrial constraint: the scarcity of data. While it is well known that more data can lead to better model performance, in many real-world applications data is limited. This competition aimed to explore whether Scientific Machine Learning (SciML) methods could still yield relevant results under such constraints.

---

[1]Airfoil simulation setting

**Multi-scale modeling and boundary layer challenges**  CFD fields are inherently multi-scale. While pressure may vary gradually in the freestream and in regions of attached flow, it can also exhibit sharp gradients near stagnation points, separation zones, and in adverse pressure recovery regions. In contrast, velocity fields exhibit steep gradients within boundary layers thin near-wall regions where viscous effects dominate and the flow transitions from zero velocity at the wall to freestream values. These gradients span multiple orders of magnitude over a distance of just a few microns and must be captured with high fidelity to ensure accurate prediction of skin friction, flow attachment, and separation.

From a turbulence modeling perspective, resolving the boundary layer is critical for drag estimation, transition prediction, and heat transfer. In RANS-based models, the accuracy of near-wall treatment depends strongly on grid resolution and turbulence closure assumptions. Surrogate models must not only regress bulk fields like pressure and velocity, but also recover their gradients with sufficient precision to represent thin viscous sublayers, which are spatially sparse and sensitive to numerical noise.

Wall shear stress, defined as the tangential component of the surface traction, is given by:

$$\tau_w = \mu \left. \frac{\partial \bar{u}_{\parallel}}{\partial n} \right|_{\text{wall}},$$

where $\mu$ is the dynamic viscosity, $n$ is the wall-normal direction, and $\bar{u}_{\parallel}$ is the tangential velocity component. Accurate estimation of $\tau_w$ requires ML surrogates to consistently approximate near-wall velocity gradients, effectively learning localized Jacobians of the flow field a substantially harder task than pointwise regression.

**From fields to forces: integration sensitivity**  Aerodynamic performance is quantified through global quantities such as lift and drag, which are not directly simulated but computed via surface integrals. Let $S$ denote the airfoil surface and $n$ the outward normal. The total force vector is given by:

$$F = \int_S \sigma \cdot n \, dS,$$

where $\sigma$ is the stress tensor, primarily driven by pressure in low-speed RANS flows. The drag ($D$) and lift ($L$) components are obtained by projection onto the flow-parallel ($u_{\parallel}$) and flow-orthogonal ($u_{\perp}$) directions:

$$\begin{cases} D = F \cdot u_{\parallel} \\ L = F \cdot u_{\perp} \end{cases}. \tag{5}$$

These coefficients, $C_D$ and $C_L$, are highly sensitive to surface pressure and shear stress distributions. Even minor inaccuracies in ML-predicted surface fields can accumulate through integration, leading to large discrepancies in force estimation. This is based on our observations during the final phase of the competition. Thus, models displaying a strong performance ought to make accurate predictions near the boundary, particularly with respect to the aerodynamic behavior. For the sake of completeness, we add that even major inaccuracies in ML-predicted surface fields could cancel out through integration, leading to small discrepancies in force estimation. Thus, the flow fields in the boundary layer might be totally incorrect, but the force could have a small error. While we acknowledge that, theoretically, such phenomena could occur, in practice this appears unlikely to be a dominant effect. looking forward, it would be valuable to also consider the integrated absolute force error in future evaluations. Doing so would provide an even more robust metric to address this potential issue

**Design space exploration**  In industrial CFD applications, it is common to evaluate thousands of design candidates across high-dimensional parametric spaces. Efficient exploration of these spaces relies on the ability of surrogates to generalize across diverse geometric configurations while maintaining predictive reliability. Importantly, surrogate models must preserve relative ranking of candidate designs, as many optimization routines rely on this property to converge toward optimal solutions.

Metrics such as Spearman's rank correlation between predicted and ground-truth aerodynamic coefficients are critical for measuring the surrogate's reliability in this setting. Poor ranking, even in the presence of low absolute error can mislead design selection processes. Moreover, surrogates must be robust to subtle geometric perturbations and maintain consistency across interpolated and extrapolated regions of the design space.

**Aerodynamic shape optimization**    The ultimate goal of many CFD surrogate modeling pipelines is to enable fast, accurate aerodynamic shape optimization. This involves differentiable pipelines in which gradients of performance metrics with respect to shape parameters can be computed efficiently and reliably. As such, ML models must support either direct differentiation or sensitivity analysis through adjoint-compatible architectures.

In this setting, high fidelity in boundary layer regions is non-negotiable: shape changes often trigger subtle shifts in separation points or adverse pressure gradients, and inaccurate modeling can lead to convergence to physically meaningless optima. ML surrogates that fail to capture such effects are of limited utility, regardless of inference speed. Consequently, industrial deployment of surrogates in shape optimization contexts requires rigorous validation, physical consistency, and close alignment with trusted CFD solvers.

**Toward industrial-ready ML surrogates**    Industrial deployment of ML surrogates demands more than computational speed. Models must generalize across geometries and flow conditions, preserve physical realism, and integrate seamlessly with CAD/CFD pipelines. Supporting variable mesh resolutions, mesh-free inference, and providing uncertainty quantification are additional enablers for production-readiness. Future iterations might also explore interpretability metrics, acknowledging both their importance in trust-sensitive engineering applications and the inherent challenge they pose for SciML methods, where balancing physical constraints with model transparency remains an open research question. While translating competition-winning models into industrial-grade solutions requires extensive engineering beyond this competition's scope, we facilitated future deployment efforts by mandating open-source submissions, enabling continued research and potential integration into real-world workflows.

## Appendix C    Score computation

We propose an *homogeneous and method-agnostic evaluation* of submitted solutions to learn the airfoil design task using the LIPS platform. The evaluation is performed using 3 categories mentioned in Section 2: ML-Related, Physics-relevant metrics & OOD generalization. For each category, specific criteria related to the airfoils design task are defined. The global score is computed based on linear combination of the scores related to three evaluation criteria categories:

$$Global\_Score = \alpha_{ML} \times \textbf{\textit{Score}}_{\textbf{ML}} + \alpha_{OOD} \times \textbf{\textit{Score}}_{\textbf{OOD}} + \alpha_{PH} \times \textbf{\textit{Score}}_{\textbf{Physics}}, \quad (6)$$

where $\alpha_{ML}$, $\alpha_{OOD}$ and $\alpha_{PH}$ are the coefficients to calibrate the relative importance of ML-Related, Application-based OOD, and Physics-relevant metrics categories respectively.

We explain in the following subsections how each of the three sub-scores were calculated in the preliminary edition. The evaluation criteria have been slightly evolved during this new edition for a better consideration of industrial applicability.

**ML-related Category score calculation**    This sub-score is calculated based on a linear combination of 2 sub-criteria: accuracy and speedup.

$$Score_{ML} = \alpha_A \times \textbf{\textit{Score}}_{\textbf{Accuracy}} + \alpha_S \times \textbf{\textit{Score}}_{\textbf{Speedup}}, \quad (7)$$

where $\alpha_A$ and $\alpha_S$ are the coefficients to calibrate the relative importance of accuracy and speedup respectively. For each quantity of interest, the accuracy sub-score is calculated based on two thresholds that are calibrated to indicate if the metric evaluated on the given quantity provides unacceptable/acceptable/great result. It corresponds to a score of 0 point / 1 point / 2 points, respectively. Within the sub-category, let: • $Nr$, the number of unacceptable results overall; • $No$, the number of acceptable results overall; • $Ng$, the number of great results overall.

Let also $N$, given by $N = Nr + No + Ng$. The score expression is given by

$$Score_{\textbf{Accuracy}} = \frac{1}{2N}(2 \times Ng + 1 \times No + 0 \times Nr) \quad (8)$$

To compute the accuracy, the Mean Squared Error (MSE) between the observations and predictions are computed, where the data are normalized (mean zero and std 1) per each Airfoil simulation

before the metric computation. This follows from the work proposed in [34] and allows to remove the scale bias and to be comparable across diverse setups (geometry, boundary conditions and flow regimes). As such, if we consider that the evaluation datasets (test and OOD) include an ensemble of $S$ simulations of the airfoil, with each simulation including an ensemble of $N$ samples, the new formulation of this metric could be written as follows:

$$MSE = \frac{1}{S} \sum_{s=1}^{S} \frac{1}{N} \sum_{i=1}^{N} (\tilde{obs}_{i,s} - \tilde{pred}_{i,s})^2,$$

where $\tilde{obs}$ and $\tilde{pred}$ represent the normalized data as mentioned previously.

Recall that it is not possible to have a better accuracy than the physical solver. The core idea behind using thresholds is that we expect a certain level of accuracy to be achieved for the physical task. Once this criterion is met, further improvements beyond that threshold do not increase the score. The thresholds were basically calibrated with a baseline to be improved to obtain a specific global score. In the first edition of this challenge, we used a fully connected neural network. For this competition, we replace the baseline with the top entry from the first edition and recalibrated the thresholds accordingly to make it even more challenging to overcome the limitation of the new baseline.

A perfect score is obtained if all the given quantities provides great results. Indeed, we would have $N = Ng$ and $Nr = No = 0$ which implies $Score_{\textbf{Accuracy}} = 1$.

While one of the primary motivations of Scientific Machine Learning (SciML) is to reduce computational time for solving PDEs, the literature, to the best of our knowledge, does not consistently define "computational time" in a manner that allows fair and systematic comparison across different configurations

Thus, For the speed-up criteria, we calibrate the score using the $log_{10}$ function by using an adequate threshold of maximum speed-up to be reached for the task, meaning

$$Score_{\textbf{Speedup}} = min\left(\left(\frac{log_{10}(SpeedUp)}{log_{10}(SpeedUpMax)}\right), 1\right), \tag{9}$$

where

- $SpeedUp$ is given by

$$SpeedUp = \frac{time_{\textbf{PhysicalSolver}}}{time_{\textbf{Inference}}}, \tag{10}$$

- $SpeedUpMax$ is the maximal speedup allowed for the airfoil use case
- $time_{\textbf{ClassicalSolver}}$: the elapsed time to solve the physical problem using the classical solver
- $time_{\textbf{Inference}}$: the inference time.

In particular, there is no advantage in providing a solution whose speed exceeds $SpeedUpMax$, as one would get the same perfect score $Score_{\textbf{Speedup}} = 1$ for a solution such that $SpeedUp = SpeedUpMax$.

Note that, while only the inference time appears explicitly in the score computation, the training time is considered via a fixed threshold: if the training time overcomes 72 hours on a single GPU, the proposed solution will be rejected. Thus, its global score is equal to zero.

Finally, we recall that the aggregation function's main purpose was to be able to provide a fair ranking of each submission. We highlight that achieving a perfect balance across such distinct objectives is not trivial at all. Thus, even though it was designed for this specific usecase and enable to take into account each solution performances across several relevant criteria with an greater weight on the most critical aspects, there is still room for improvement regarding its design.

**Physical consistency score calculation**   While the machine learning metrics are relatively standard, the physical metrics are closely related to the underlying use case and physical problem. There are two physical quantities considered in this challenge namely: the drag and lift coefficients. For each of them, we compute two coefficients between the observations and predictions: • The spearman correlation, a nonparametric measure of the monotonicity of the relationship between two datasets

(Spearman-correlation-drag : $\rho_D$ and Spearman-correlation-lift : $\rho_L$); • The mean relative error (Mean-relative-drag : $C_D$ and Mean-relative-lift : $C_L$).

For the physical consistency sub-score, we evaluate the relative errors of physical variables. For each criteria, the score is also calibrated based on 2 thresholds and gives 0 /1 / 2 points, similarly to $score_{\textbf{Accuracy}}$, depending on the result provided by the metric considered.

**OOD generalization score calculation**   This sub-score will evaluate the capability on the learned model to predict OOD dataset. In the OOD testset, the input data are from a different distribution than those used for training. the computation of this sub-score is similar to $score_{ML}$ and is also based on two sub-criteria: accuracy and speed-up. To compute accuracy we consider the criteria used to compute the accuracy in $score_{ML}$ in addition to those considered in physical consistency.

**Score Calculation Example**   To demonstrate the calculation of the everall score, we utilize the notation established in the preceding section. We provide in table 3 several examples for the score calculation based on the results obtained in the previous edition of the current competition, including the top 5 solutions. We illustrate in the following how to calculate the score of the baseline (Fully connected) based on the parameters used in the preliminary edition: $\alpha_{ML} = 0.4$, $\alpha_{OOD} = 0.3$, $\alpha_{PH} = 0.3$, $\alpha_A = 0.75$, $\alpha_S = 0.25$, $SpeedUpMax = 10000$.

- $Score_{\textbf{ML}} = 0.75 \times (\frac{2 \times 1 + 1 \times 1 + 0 \times 3}{2 \times 5}) + 0.25 \times \frac{log_{10}(750)}{log_{10}(10000)} \approx 0.405$

- $Score_{\textbf{OOD}} = 0.75 \times (\frac{2 \times 1 + 1 \times 1 + 0 \times 7}{2 \times 9}) + 0.25 \times \frac{log_{10}(750)}{log_{10}(10000)} \approx 0.305$

- $Score_{\textbf{Physics}} = (\frac{2 \times 1}{2 \times 4}) = 0.25$

For accuracy scores, the detailed results with their corresponding points are reported in Table 4. Speed-up scores are calculated using the equation 10 as follows:

- $time_{\textbf{PhysicalSolver}} = 1500s$, $time_{\textbf{Inference-ML}} = 2s$ , $time_{\textbf{Inference-OOD}} = 2s$
- $Score_{\textbf{SpeedupML}} = Score_{\textbf{SpeedupOOD}} = \frac{1500}{2} = 750$

Then, by combining them, the global score is $Score_{FC} = 0.4 \times 0.405 + 0.3 \times 0.305 + 0.3 \times 0.25 = 0.3285$, therefore 32.85%.

Table 3: Scoring table of the preliminary edition of the competition [35] for Airfoil design task under 3 categories of evaluation criteria. The performances are reported using three colors computed on the basis of two thresholds. Colors meaning: 🔴 Unacceptable (0 point) 🟠 Acceptable (1 point) 🟢 Great (2 points). Reported results: Baseline solution (Fully Connected NN), OpenFOAM (Ground Truth), and the 5 winning solutions from the preliminary edition of the Competition. MMGP : Mesh morphing Gaussian Process. GGN-FC: a combined Graph Neural network (GNN) and FC appraoch. MINR: Multiscale Implicit Neural Representations. Bi-Trans : Subsampled bi-transformer. NeurEco : NeurEco based MLP.

| | | ML-related (40%) | | Physics (30%) | OOD generalization (30%) | | | Global Score (%) |
|---|---|---|---|---|---|---|---|---|
| | Method | Accuracy(75%) | Speed-up(25%) | Physical Criteria | OOD Accuracy(42%) | OOD Physics(33%) | Speed-up(25%) | |
| | | $\overline{u}_x$ $\overline{u}_y$ $\overline{p}$ $\overline{v}_t$ $\overline{p}_s$ | | $C_D$ $C_L$ $\rho_D$ $\rho_L$ | $\overline{u}_x$ $\overline{u}_y$ $\overline{p}$ $\overline{v}_t$ $\overline{p}_s$ | $C_D$ $C_L$ $\rho_D$ $\rho_L$ | | |
| | OpenFOAM | 🟢🟢🟢🟢🟢 | 1 | 🟢🟢🟢🟢 | 🟢🟢🟢🟢🟢 | 🟢🟢🟢🟢 | 1 | **82.5** |
| | Baseline(FC) | 🟠🟠🟠🟢🔴 | 750 | 🔴🔴🟠🔴 | 🟠🟠🟠🟢🔴 | 🔴🔴🟠🔴 | 750 | **32.85** |
| **Rank** | | Preliminary Edition : Top 5 solutions | | | | | | |
| 1 | MMGP [22] | 🟢🟢🟢🟢🟢 | 27.40 | 🟢🟢🟠🟢 | 🟢🟢🟢🟢🟢 | 🟢🟢🟠🟢 | 28.08 | **81.29** |
| 2 | GNN-FC | 🟢🟢🔴🟢🟠 | 570.77 | 🟢🟠🟠🟢 | 🟠🟠🔴🟠🔴 | 🟢🟠🟠🟠 | 572.3 | **66.81** |
| 3 | MINR | 🟢🟢🟠🟢🟠 | 518.58 | 🟠🟠🔴🟢 | 🟢🟢🟠🟢🔴 | 🟠🟠🟢🟠 | 519.21 | **58.37** |
| 4 | Bi-Trans | 🟢🟢🟠🟢🔴 | 552.97 | 🟠🟠🔴🟢 | 🟠🟠🟠🟢🔴 | 🟢🟠🔴🟠 | 556.46 | **51.24** |
| 5 | NeurEco | 🟢🟢🟢🟢🔴 | 44.93 | 🟠🟠🟠🟢 | 🟢🟢🟠🟢🔴 | 🟠🟠🔴🟢 | 44.78 | **50.72** |

# Appendix D   Additional details on Competition Design and Organizational Insights

## D.1   Competition phases

The competition was organized through three phases

Table 4: Accuracy scores calculation of the FC solution. $T1$ designates the first threshold below which the performance is great (green color) and $T2$ designates the second threshold above which the error is not acceptable (red). Specific thresholds are considered for each criteria in different categories.

| Category | Criteria | obtained results | Thresholds | min/max | obtained score |
|---|---|---|---|---|---|
| ML-Related | $\overline{u}_x$ | 0.208965 | T1=0.1 / T2 =0.2 | min | 🔴 0 point |
| | $\overline{u}_y$ | 0.144508 | T1=0.1 / T2=0.2 | min | 🟠 1 point |
| | $\overline{p}$ | 0.193066 | T1=0.02 / T2=0.1 | min | 🔴 0 point |
| | $\overline{\nu}_t$ | 0.277285 | T1=0.5 / T2=1.0 | min | 🟢 2 points |
| | $\overline{p}_s$ | 0.425576 | T1=0.08 / T2 =0.2 | min | 🔴 0 point |
| | | | | | $N = 5,\ Nr = 3,\ No = 1,\ Ng = 1.$ |
| physical consistency | $C_D$ | 16.345740 | T1=1 / T2 =10 | min | 🔴 0 point |
| | $C_L$ | 0.365903 | T1=0.2 / T2 =0.5 | min | 🟠 1 point |
| | $\rho_D$ | -0.043079 | T1=0.5 / T2 =0.8 | max | 🔴 0 point |
| | $\rho_L$ | 0.957070 | T1=0.94 / T2 =0.98 | max | 🟠 1 point |
| | | | | | $N = 4,\ Nr = 2,\ No = 2,\ Ng = 1.$ |
| OOD Generalization | $\overline{u}_x$ | 0.322766 | T1=0.1 / T2 =0.2 | min | 🔴 0 point |
| | $\overline{u}_y$ | 0.199635 | T1=0.1 / T2=0.2 | min | 🟠 1 point |
| | $\overline{p}$ | 0.333169 | T1=0.02 / T2=0.1 | min | 🔴 0 point |
| | $\overline{\nu}_t$ | 0.431288 | T1=0.5 / T2=1.0 | min | 🟢 2 points |
| | $\overline{p}_s$ | 0.805426 | T1=0.08 / T2 =0.2 | min | 🔴 0 point |
| | $C_D$ | 21.793367 | T1=1 / T2 =10 | min | 🔴 0 point |
| | $C_L$ | 0.711271 | T1=0.2 / T2 =0.5 | min | 🔴 0 point |
| | $\rho_D$ | -0.043979 | T1=0.5 / T2 =0.8 | max | 🔴 0 point |
| | $\rho_L$ | 0.917206 | T1=0.94 / T2 =0.98 | max | 🔴 0 point |
| | | | | | $N = 9,\ Nr = 7,\ No = 1,\ Ng = 1.$ |

- *Warm-up phase (5 weeks).* In this phase, participants get familiarized with provided material and the competition platform, made their first submissions and provided feedback to organizers, based on which some adjustments were made for the development phase.

- *Development phase (10 weeks).* During this phase, participants developed their solutions. By submitting them on the evaluation pipeline provided through Codabench, they were able to obtain a score and global ranking in the leaderboard and to test their already trained models against a provided validation dataset. They can also have access continuously to the global score corresponding the submitted solution.

- *Final phase (2 week).* The organizers prepared the final ranking by executing each submission multiple times and reported official results using means and standard deviations. The winners were announced during the NeurIPS 2024 conference.

## D.2 Submissions statistics

Figure 5 presents the activity level of the participants during the different phases of the competition. Two high peak activity periods could be observed at the second part of the development phase and also during the extended period. During the warm-up and the beginning of the development phase, the participants started to familiarize with the usecase and the competition procedure and we had a high stake discussion period in the dedicated discussion channel. It shed light also on the importance of anticipating an extension at the end of the competitions to allow the the last minutes submissions and adjustments.

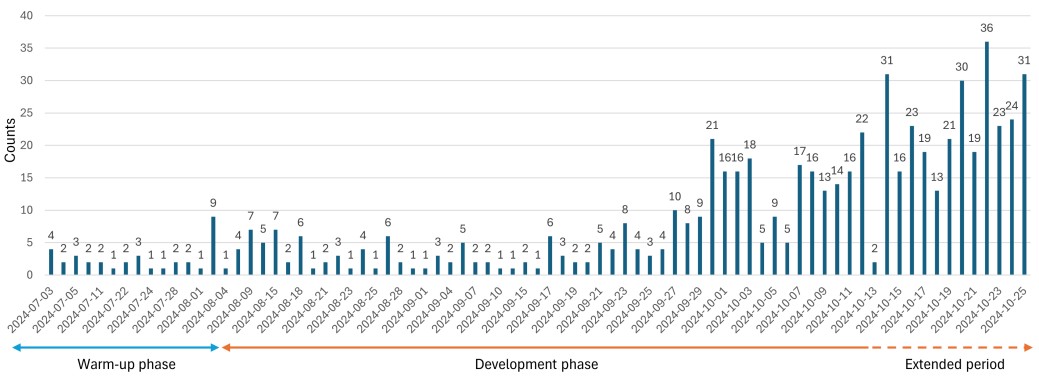

Figure 5: Submission statistics during the different phases of the competition

### D.3 Competition leaderboard

Table 5 presents the complete competition leaderboard. The participants are presented with their pseudo-names in this table. The rankings were determined based on a global score that aggregated performance across several key categories, including machine learning (accuracy and speed-up), physics, and out-of-distribution (OOD) generalization. The top-ranked solution outperformed the physical baseline (the green line), while the top nine submissions exceeded the machine learning baseline, which had been established as both a reference starting point and a threshold for solution acceptance (red line).

Table 5: Competition leaderboard with all the participants. A more detailed analysis of winning solutions is provided in Table 1. The green line indicates the physical solver threshold and the red line the acceptance threshold. The participants are presented with their pseudo-names in this table.

|  |  | | Criteria categories | | |
|------|-------------|--------------|------|---------|------|
| Rank | Participant | Global score | ML | Physics | OOD |
| 1 | Safran-Tech | 84.68 | 0.89 | 0.88 | 0.76 |
| 2 | aK | 80.54 | 0.86 | 0.88 | 0.66 |
| 3 | adama | 77.45 | 0.76 | 0.88 | 0.7 |
| 4 | aeroairbus | 71.21 | 0.77 | 0.75 | 0.59 |
| 5 | jacobhelwig | 50.61 | 0.54 | 0.5 | 0.46 |
| 6 | optuq | 49.25 | 0.59 | 0.38 | 0.48 |
| 7 | doomduke2 | 44.81 | 0.52 | 0.5 | 0.31 |
| 8 | shyamss | 44.2 | 0.47 | 0.5 | 0.34 |
| 9 | Xiaoli | 39.22 | 0.46 | 0.38 | 0.32 |
| 10 | epiti | 17.57 | 0.18 | 0.12 | 0.22 |

### D.4 Incentive measures

As has already been mentioned, during the different phases of the competition, the participants had access to the provided computation resources allowing them to train and evaluate their solutions via Codabench interface. We have also imposed that the final submissions to be trained on our servers. These considerations encouraged the participants who have not access to complex resources (*e.g.*, students), to be able to participate in to the competition and to be evaluated with respect to the same standardized and fair evaluation protocol.

To familiarize the participants with the competition problem, tools and resources, we have provided some materials and organized a webinar. The provided starting kit included a set of self-explanatory Jupyter notebook, demonstrating the different steps of the competition from importing and pre-processing the dataset to submission on Codabench. We have also provided a communication channel via Discord, allowing to debug the participant's problem during the different phases of the competition. Finally, for a more interactive session with the participants, we have organized a webinar during the first month.

Various prizes were also considered to encourage the more dynamic competition between the participants. In addition to the main prizes attributed to the first three places (4,000€, 2,000€ and 1,000€), a special prize (1,000€) was also considered for the best student solution which has been affected to the fourth ranked solution proposed by a student (see Table 1).

**Effectiveness of competition materials**    The starting kit was specifically designed to help participants with no background in computational physics by formalizing the underlying problem as a classical machine learning task. Also, as we have provided active support throughout the competition, candidates were able to call for assistance easily if required. As far as we can tell, most of the questions asked were not related to the understanding of the competition materials. Thus, it is reasonable to assume it was not an issue for the participants.

**Use of provided baselines by participants**    Several participants have used the provided baselines. For some of them, it was a convenient way to compare the performances of their approach to a reference. For others, some baselines were a mere starting point. For instance, in section 4.4, they were carefully analyzed and improved step by step.

**Impact of multi-criteria evaluation on hybrid model usage**    In this competition, and quite often in SciML in general, there is a trade-off between accuracy and speed-up. The ML approaches are allegedly faster than classical physical solvers but are also known to be less accurate, as shown with the ML baselines provided. As the multi-criteria evaluation takes into account both aspects, obtaining an acceptable result strongly encourages candidates to use hybrid models by providing a more comprehensive and robust assessment across multiple important aspects, such as accuracy, physical consistency, and efficiency.

## Appendix E    Extended discussion

### E.1    Suggestions for Enhancing the Evaluation Protocol

We provide in this section some details about the limitations regarding the evaluation procedure we used during the competition. The speed-up metric that we used is:

$$s = \frac{Time_{ClassicalSolver} + Time_{evaluation}}{Time_{Inference} + Time_{evaluation}} \tag{11}$$

This design choice aligns with Amdahl's law, which states that the overall speed-up of a system is constrained by the proportion of time spent on non-accelerated components. For example, if evaluation represents 1% of the total pipeline runtime, the theoretical maximum speed-up is limited to 100×. Thus, as inference becomes faster, evaluation time increasingly becomes the dominant bottleneck.

Importantly, we verified that including or excluding evaluation time in the speed-up metric did not alter the final ranking of submissions, which supports the robustness of our metric under these constraints.

While the current evaluation protocol provides a reasonable basis for comparison, it could be improved to better reflect the complexity of comparing ML-based solvers with classical numerical methods in practical settings. Several directions could be explored to refine future benchmarking methodologies. One avenue for refinement involves accounting for the effective acceleration of ML models, as proposed in the literature [33]. A representative formulation includes both one-time and amortized costs:

$$C_{data} + C_{train} + N\frac{t_b}{s} = Nt_b \tag{12}$$

with

- $C_{data}$: data generation time
- $C_{train}$: training time
- $N$: number of samples for evaluation of ML

- $t_b$: average reference time

- $s$: average speed-up

This equation defines the break-even point at which the ML approach becomes computationally favorable. The smaller the required $N$, the more advantageous the ML method is in practice. In our setting, generating the training data required approximately 43 hours (with each OpenFOAM simulation taking 25 minutes), and we allowed up to 72 hours of training during the development phase. These costs, which are absent in traditional solver pipelines, are significant and should be incorporated in future evaluations to reflect real-world deployment costs more accurately. Additionally, future benchmarking efforts should consider normalization by accuracy or runtime, comparing methods either at equivalent predictive accuracy or under fixed computational budgets. While this introduces practical and methodological challenges, it would enable a more interpretable assessment of trade-offs between performance and efficiency.

Another key consideration is hardware fairness. Classical solvers typically run on CPUs, while ML-based solutions benefit from parallelism and specialized acceleration on GPUs. In this competition, we aimed to ensure fairness by matching high-performance CPUs and GPUs appropriately. However, future benchmarks could benefit from more systematic treatment of hardware performance variability, potentially standardizing compute budgets or energy usage across approaches.

Finally, extending evaluation protocols to include generalization to unseen geometries, boundary conditions, or mesh types (as well as robustness to perturbations or noise) would make them more representative of practical deployment scenarios. Such considerations are essential as ML models move beyond proof-of-concept demonstrations and toward integration into high-stakes scientific and engineering workflows. While factors such as deployment cost, model size, memory usage, and inference latency can significantly affect practical utility, they were not included in our evaluation. In a competition setting, clear and consistent ranking is essential, and incorporating these additional factors would have required complex adjustments to the scoring system.

## E.2 Recommendations for designing new competitions in SciML

The design of machine learning competitions in scientific computing (SciML) involves several non-trivial choices that can significantly impact the relevance and fairness of the evaluation. Based on our experience, we outline a set of recommendations to guide the design of future challenges.

**Selection of the Physical Problem** The first step is to clearly define a physically meaningful task and associated evaluation criteria grounded in domain knowledge. This includes selecting a representative partial differential equation (PDE) and establishing a reference solution based on state-of-the-art numerical methods. The choice of the baseline should be well-justified, both in terms of numerical accuracy and computational efficiency. In this context, collaboration with domain experts is highly recommended to ensure the physical and numerical validity of the setup. To improve transparency and mitigate the risks of misleading comparisons—as highlighted in [36]—metadata about the baseline solver (e.g., time discretization, interpolation schemes, mesh resolution) should be systematically provided. Additionally, offering multiple baselines, including both standard numerical solvers and ML-based alternatives, can better contextualize performance.

**Dataset Design and Generalization** Since SciML tasks typically rely on data generated from simulations, it is important to avoid overexposure of specific datasets. Excessive tuning to a single, publicly available dataset can lead to over-specialization, limiting generalizability. If the goal is to promote models with broader applicability, evaluation should span multiple physical configurations or PDE problems. In our case, we were constrained to use a public dataset for evaluation. While this approach ensures transparency and reproducibility, it also introduces the risk of model overfitting to known test cases. To address this, the use of hidden or held-out datasets—particularly introduced during a post-competition phase—can help assess the generalization capabilities of submitted models more effectively. This becomes especially important for competitions organized over multiple seasons, where participants may iteratively adapt their models based on previously released test cases. Introducing new, unseen evaluation data in later phases or seasons is therefore critical to maintaining a meaningful and unbiased benchmark.

**Applicability to existing dataset**   In this section, we provide a brief assessment of existing physics-based datasets that could serve as candidate foundations for a SciML competition. AhmedML [37] is a high-fidelity, transient, scale-resolving CFD dataset for the widely used Ahmed car body with geometric variability. Such datasets are valuable for testing whether ML-augmented CFD models scale beyond academic toy cases, as performance and stability may degrade in realistic 3D regimes despite success on small 2D meshes. The dataset includes solver settings and PDE-derived quantities (e.g., drag, lift, wall shear stress). While suitable in principle for model development, its size raises operational constraints for a competition: organizers must bear the cost of training and handling submissions at scale. Moreover, a standardized post-processing pipeline (analogous to AirfRANS) is needed to compute evaluation quantities from ML outputs in a physically consistent manner. The Well [38] aggregates large-scale spatiotemporal simulations across 16 physical domains with many boundary conditions and transient trajectories. This scale enables benchmarking physics-agnostic generalization. ML baselines and a PyTorch interface are provided, but a library-agnostic API would improve accessibility. Although diverse metrics (spatial, spectral, data-only) are available, no dataset-specific physical metric is prescribed. For evaluating physical fidelity, generic PDE-residual metrics or similar physics-aware criteria are likely necessary. Given the autoregressive nature of many tasks, solver metadata (schemes, stability, etc.) should also be exposed for interpretability. Finally, PDEBench [39] offers an integrated SciML benchmark suite with multiple datasets, baselines, and both ML and physics metrics in 1D–3D. Nevertheless, designing a competition requires a principled scoring function, which we discuss in the next section.

**Scoring Function and Evaluation Robustness**   The design of the scoring metric is critical, as it guides model optimization and participant behavior. The scoring function should align with the competition objectives, for example by balancing solution accuracy with inference speed. It is essential to ensure that unrealistic solutions (e.g., extremely fast but physically inaccurate) receive appropriately low scores. Robustness of the scoring scheme should be validated across a range of scenarios to ensure consistency. Moreover, evaluations should not rely solely on average performance: variability across simulations should be reflected in the overall score to capture model stability. This is particularly important when metrics exhibit high variance due to stochastic elements in ML models.

**Statistical Validation and Post-Competition Evaluation**   In the post-competition phase, we re-evaluated each submission multiple times (7 experiments) and applied statistical testing to account for the stochasticity inherent in ML approaches. This helped ensure that the final rankings were not unduly affected by random fluctuations.

While integrating such statistical validation into the development phase would be ideal, it may significantly increase evaluation costs and slow down feedback loops. Therefore, we recommend reserving rigorous statistical analysis for the final evaluation phase, while maintaining a more lightweight approach during development.

## Appendix F    MMGP: additional details and results

### F.1    Data analysis

We analyze the data to try to identify any outlier in the training set. After plotting the drag coefficient with respect to the lift coefficient of the training set in Figure 6, we notice than one configuration is very different from the trend formed by all the other ones. This configuration may differ for various reasons (out of the distribution of the others, or maybe the CFD computation is not converged), and we decide to remove it from the training set.

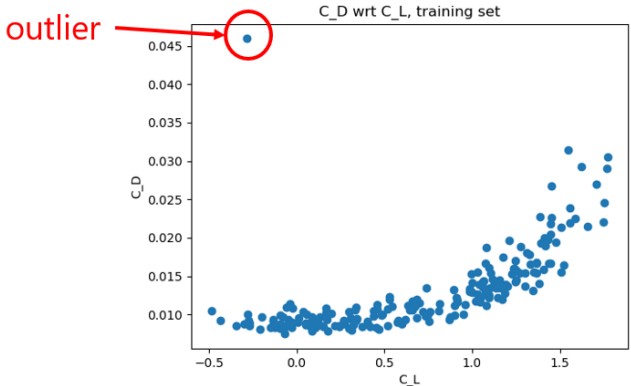

Figure 6: Outlier identifier in the training set by representing the drag coefficient with respect to the lift coefficient.

### F.2    Dedicated pretreatments

The first pretreatment of MMGP is the morphing of each mesh onto a common shape, see the workflow in Figure 1. In the AirfRANS dataset, the meshes are 2D, but their external boundary is very irregular, with variable number of "peaks", which can lead to a additional difficulties when deriving a deterministic morphing algorithm. Hence, we pretreat the sample to extend each mesh up to a bounding box common to each samples, see Figure 7. We use Muscat [40, 41], a library developed at Safran, for all mesh and field manipulations

As common shape, we choose the geometry corresponding to ont of the pretreated mesh of the training set. As morphing algorithm, we used a Radial Basis Function (RBF) morphing, which requires the transformation of some control points and computes the transformation of all the other ones. The choice and transformation of the control points is illustrated in Figure 8.

The morphing of the second mesh of the training set is illustrated in Figure 9.

### F.3    Dimensionality reduction

All the fields (input coordinates and output fields) are projected on the common mesh: the pretreated and morphed mesh of the first sample of the training set. For the dimensionaly reduction, we use the snapshot Proper Orthogonal Decomposition (POD), see [22, Annex C]. Objects uncompressed from a reduced representation constructed by the snapshot-POD have the property of being expressed as a linear combination of the compressed samples. Hence, all zero linear relations are conserved by the reduced objects. In the AirfRANS dataset, the velocity field has a zero boundary condition at the surface of the airfoil. The scoring function of the challenge involves scalar quantities obtained by integrating fields at the surface of the airfoil, hence respecting the zero boundary condition is of paramount importance. Using the snapshot POD with MMGP, all the predicted fields are expressed as linear combinations of velocity fields, hence will automatically almost respect the zero boundary condition (since the property is verified only on the common mesh). In our solution, we use 16 modes for the shape embedding, 13 for the velocity, 24 for the pressure fields, and 12 for the turbulent viscosity.

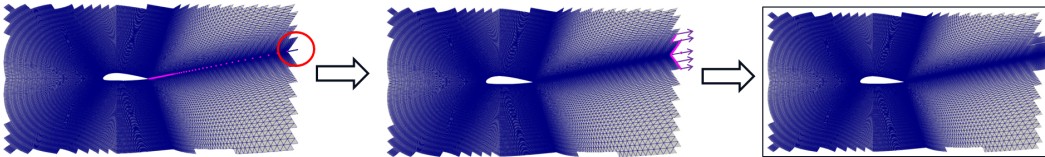

We select nodes on the external boundary close to the center of the wake, and project them onto the bounding box, in the direction of the wake. The extreme anisotropy of the mesh requires following the direction of the wake to prevent returning any element.

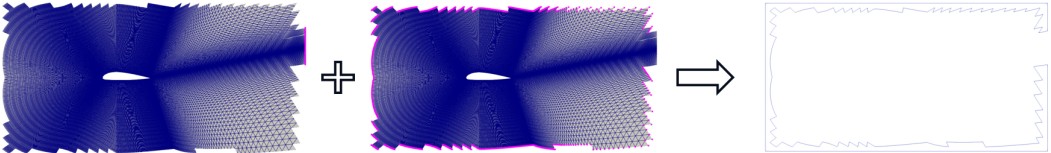

We select the remaining points on the external boundary, and construct a 1D closed mesh with these points and some chosen sampling of the remaining of the bounding box.

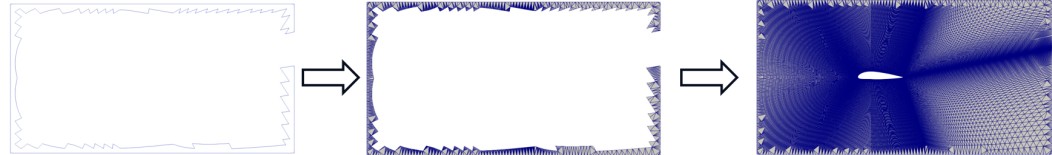

We applied a constrained Delaunay triangulation to mesh the interior of this 1D closed mesh, while keeping all the points of this 1D mesh, to obtain a 2D mesh. This mesh is fused with the input mesh to obtain the desired result. A clamped finite element extrapolation is used to extend all fields of interest onto this obtained mesh.

Figure 7: Pretreatment to extend meshes up to a common bounding box.

## F.4 Gaussian process regression

We have pretreated all the meshes, morphed them, projected the fields onto a common mesh, and reduced the dimension of the input coordinate fields and output fields. For the challenge, we have trained independent Gaussian Processes for each generalized coordinate associate to the snapshot-POD of each output field of interest. In Figure 10, we illustrate the workflow of the machine learning part of MMGP, where the used scalars are indicated.

For the Gaussian process regressor, we have re-implemented a model in pytorch, executed on CPU. We use a constant times RBF_kernel plus white noise kernel. The only hyperparameters to fit are the constant (dim=1), the lengthscales of the RBF kernel (dim=input dim) and the noise scale (dim=1), all initialized to 1 – hence, the optimization does not profit from multi-start. We use a AdamW optimizer with 1000 steps, with an adaptive step size (ExponentialLR scheduler: gamma=0.9).

## F.5 Additional improvements and results

We use various additional tricks to improve accuracy and seepdup, among them:

- the first velocity generalized coefficients are added as inputs of the pressure regressor: in the incompressible Navier-Stokes equations, the pressure acts as the Lagrange multiplier enforcing the incompressibility condition in the optimization problem on the velocity,

- Fast Morphing: we removed some checks from the RBF morphing function in Muscat [40],

- Fast Interpolation: we developed the following heuristic dedicated to linear triangles, see Figure 11:

  - find the 100 closest input triangle barycenters for each output point, and keep the input triangle containing the output point,

  - for each point with no found triangle, find the triangle barycenters closer than 0.3, then choose the barycenter with highest smaller barycentric coordinate,

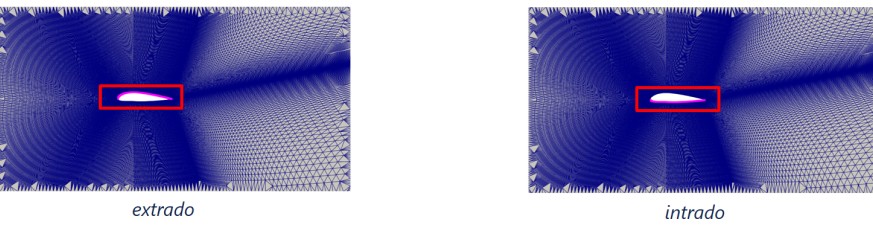

*extrado*                                    *intrado*

The intrado and the extrado points of the airfoil mesh are chosen, and carefully morphed onto the intrado and extrado of the first pretreated mesh of the training. To do that, we start by the leading edge, and we resample the 1D meshes corresponding to the intrado and the extrado the first pretreated mesh, by conserving the curvilinear abscissa of the points of the mesh to morph. Hence, we are robust to any potential change of mesh resolutions through the samples.

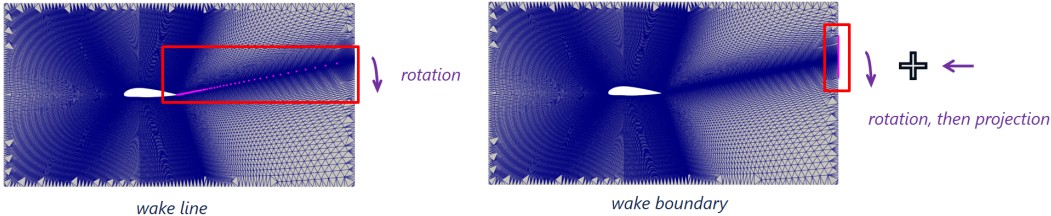

*wake line*                                  *wake boundary*

Figure 8: Transformation of the control points for the Radial Basis Function morphing.

We identify the points at the center of the wake, except the one on the boundary, and apply a rotation to align them onto an horizontal line. Then, the right boundary condition must be transformed in a way that the points stay on the boundary box, but the element are stretched above the wake, and compressed below (in the configuration of the figure). The points close to the wake being extremely close, we prefer imposing the same rotation as the wake (and reprojecting them onto the boundary box), and we apply the stretching/compression to the other points of the boundary. The points of the top, bottom and left boundary are kept fixed.

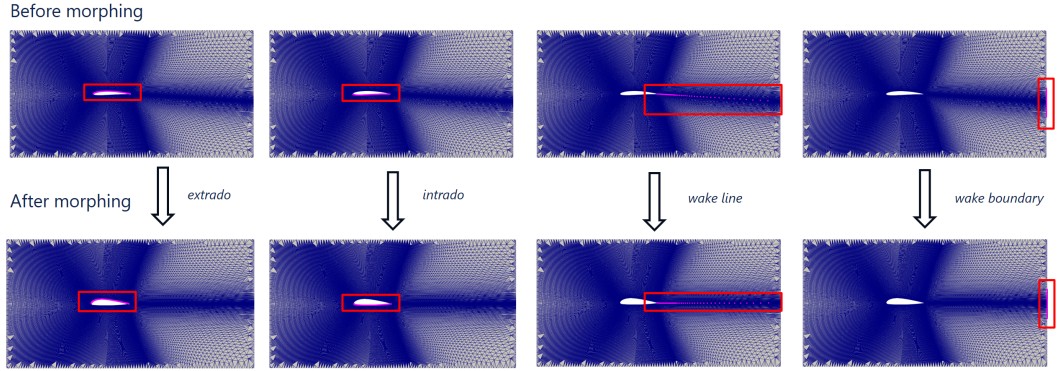

Figure 9: Morphing of the second mesh of the training set.

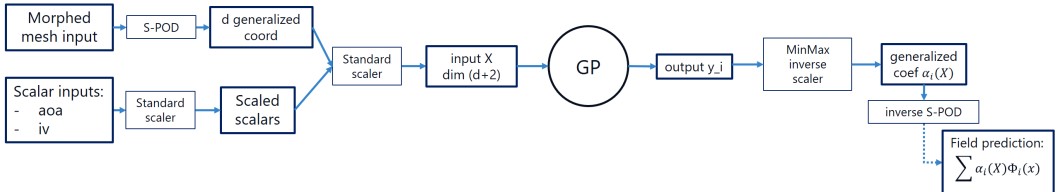

Figure 10: Workflow of the machine learning part of MMGP

– compute interpolations combining input field values with chosen triangle barycentric coordinates.

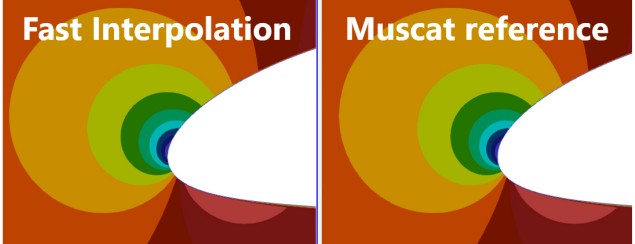

Figure 11: Fast interpolation

- We learn the angle of the wake instead of relying on mesh cell ids using a Gaussian process regressor. We take as inputs the angle of attack, the inlet velocity, and the curvilinear abscissa of intrado and extrado projected on common discretization, see Figure 12.

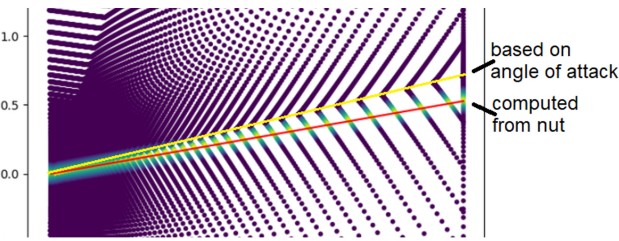

Figure 12: Wake angle learning

Some illustrations of the predicted fields are provided in Figures 13-14.

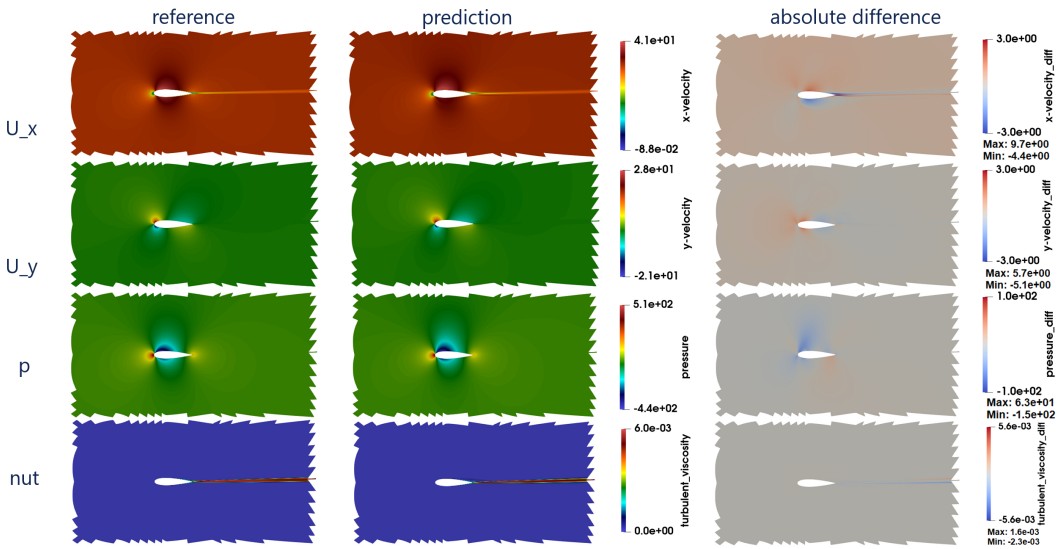

Figure 13: Illustration of the result on the first sample of the test set.

## F.6 Discussion and prospects

MMGP benefits from important advantages. It can easily handle very large meshes, can be trained on CPU, is interpretable, shows high accuracy in our experiments and on the competition use case, and comes with readily available predictive uncertainties, see [22]. With this competition, we wanted to illustrate that when problems are in small intrinsic dimension (which is the case for many industrial design settings), small models can be very efficient if they are correctly reparametrized. In the case of

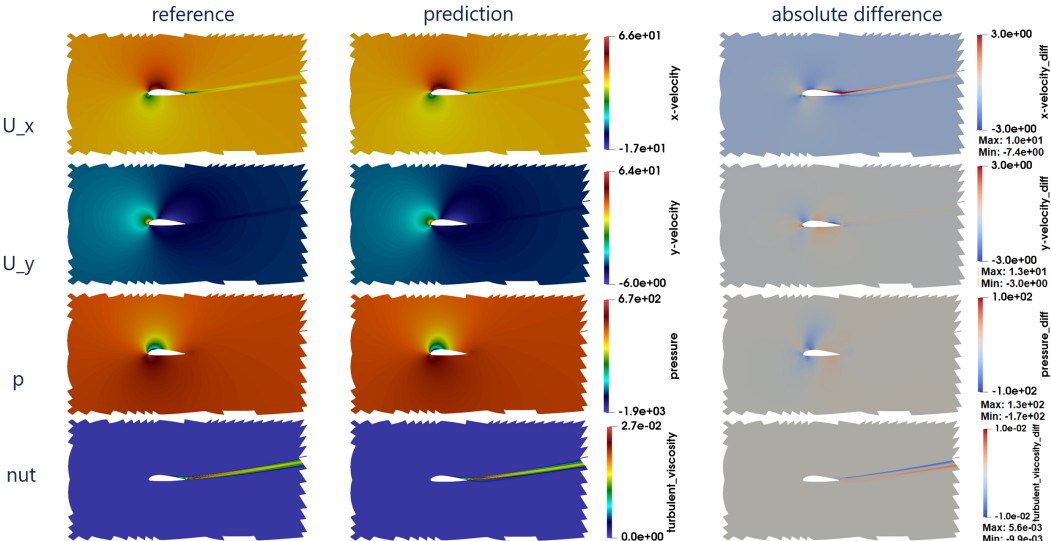

Figure 14: Illustration of the result on the first sample of the test_ood set.

MMGP, the morphing allows for this effective reparametrization. All these deterministic pretreatments aim at relieving the machine learning part of the complicated shape and field embeddings of large and variable dimension objects. We chose the simplest possible linear dimensional reduction and non-linear regression tools, produced monolithic predictions (single spatial POD decomposition for each field, no spatial partitioning), did not resort to deep learning, feature engineering, data augmentation or hybridation with physics. Furthermore, the complete learning stage (including all pretreatments) took less than 10 minutes in the competition server, without even using the provided GPU.

The main drawbacks of MMGP concern the morphing, which must still be tailored to each new use case, the assumptions of fixed topology and the limited speedup due to the requirement of computing morphing and finite element interpolation in the inference stage. These drawbacks have been recently addressed in [42, 43] where automatic shape-alignment and online-efficient morphing strategies have been developed, and in [44] where an optimization algorithm is used to derive morphings optimized to maximize the compression onto the PCA modes. We also mention that a future release of Muscat will include an extremely efficient finite element interpolation routine, that can be executed on GPU, which will drastically accelerate the remaining time-consuming step in the inference of MMGP.

## Appendix G    offset-based additional results

Some illustrations of the predicted fields are provided in Figures 15-16.

## Appendix H    MARIO: additional details and results

### H.1    Implementation Details

MARIO was implemented in `PyTorch` [45] with a modular architecture that combines coordinate-based MLPs with conditional hypernetworks. Input features are spatial coordinates, distance function, processed normals, and boundary layer mask, while the outputs are velocity, pressure, and turbulent viscosity fields. For conditioning, we used inlet velocity and airfoil geometric metrics (thickness and camber distributions sampled at 10 points along the airfoil chord). All input and conditioning features are normalized using a centered min-max scaler, while output features used standard scaling. The main network consists of 6 layers with 256 hidden units each, processing inputs through multiscale Fourier feature embeddings with scales $\{0.5, 1, 1.5\}$ to capture different frequency components of the flow field. The hypernetwork used for conditional modulation has 4 layers with 256 hidden units. During training, we randomly sample 16,000 points per airfoil to create mini-batches of

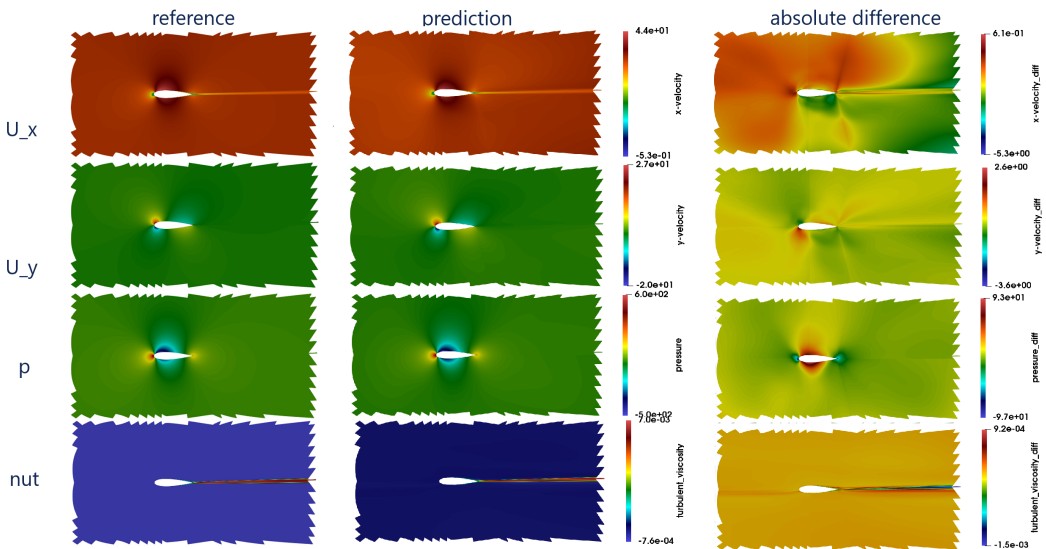

Figure 15: Illustration of the result on the first sample of the test set for the offset-based approach.

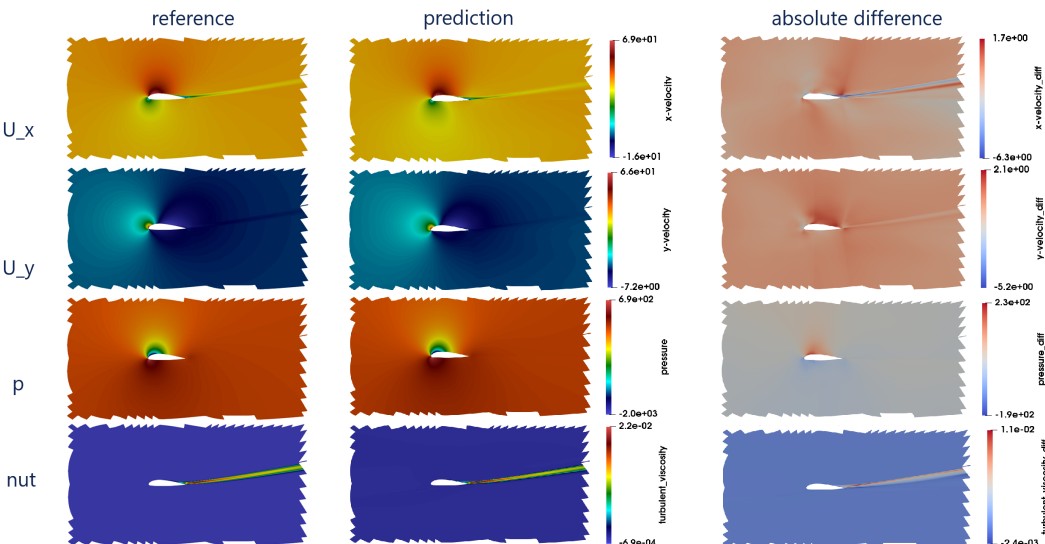

Figure 16: Illustration of the result on the first sample of the test_ood set for the offset-based approach.

size 4. The model was optimized using `AdamW` [46] with an initial learning rate of 0.001 and a `ReduceLROnPlateau` scheduler that decreased the learning rate by a factor of 0.8 after 10 epochs without improvement. To ensure stable training, we apply gradient clipping with a maximum norm of 1. The hyperparameters choices are summarized in Table 6.

## H.2 Feature Engineering

Given the limited training dataset of only 100 samples, incorporating domain knowledge through feature engineering proved crucial for achieving high predictive accuracy. Rather than relying solely on raw mesh coordinates and signed distance fields, we developed additional features that enhance the model's understanding of the underlying flow physics: volumetric normals and boundary layer masking. Volumetric normals extend surface normal information throughout the domain, providing directional guidance about each point's relationship to the airfoil surface. The boundary layer mask highlights the critical near-wall region where the most significant flow gradients occur. As

Table 6: MARIO model hyperparameters and training configuration

| Model Architecture | | Training Configuration | |
|---|---|---|---|
| Main network depth | 6 | Optimizer | AdamW |
| Main network width | 256 | Learning rate | 0.001 |
| Hypernetwork depth | 4 | Weight decay | 0 |
| Hypernetwork width | 256 | Batch size | 4 |
| Fourier feature frequencies | 128 | Points per sample | 16,000 |
| Fourier scales | $\{0.5, 1, 1.5\}$ | LR scheduler factor | 0.8 |
| | | LR scheduler patience | 10 epochs |
| | | Gradient clipping norm | 1.0 |
| | | Loss function | MSE |

demonstrated in our experiments, these features significantly accelerated convergence and improved prediction accuracy, especially in flow regions with steep gradients.

### H.2.1 Volumetric normals

Volumetric normals extend the concept of surface normals to the entire fluid domain, providing directional information about how each point relates to the airfoil geometry. These normals create a continuous vector field that helps the network understand spatial relationships between flow points and the airfoil surface.

For each point in the domain, we compute the volumetric normal as follows:

1. For each point $\mathbf{p}_i$ in the domain, we identify the closest point $\mathbf{s}_i$ on the airfoil surface.

2. We compute the vector from the domain point to the surface point: $\mathbf{v}_i = \mathbf{s}_i - \mathbf{p}_i$.

3. We normalize this vector to get the unit normal: $\mathbf{n}_i = \frac{\mathbf{v}_i}{||\mathbf{v}_i||}$.

4. To create a smooth transition from the airfoil surface to the far field, we apply a fade factor based on the normalized distance function:

$$\tilde{d}_i = \frac{d_i - d_{\min}}{d_{\max} - d_{\min}} \tag{13}$$

5. The final volumetric normal is computed as:

$$\mathbf{n}_i^{\text{proc}} = \mathbf{n}_i \cdot (d_{\max}^{\text{norm}} - d_i^{\text{norm}}) \tag{14}$$

Figure 17a shows the processed normals field, visualizing the x and y components across the domain. The impact of incorporating volumetric normals on model performance is demonstrated in Figure 17b, where we observe a substantial reduction in validation loss compared to the baseline without this feature.

### H.2.2 Boundary Layer Masking

The boundary layer is the region near the airfoil surface where the most significant gradients in flow variables occur. To help the model focus on this critical region, we developed a boundary layer mask that highlights the near-wall region with a smooth decay function.

The boundary layer mask is computed using the signed distance function (SDF) as follows:

1. We invert and normalize the distance function:

$$\hat{d} = \frac{d_{\max} - d}{d_{\max}} \tag{15}$$

2. We apply a thresholding function to create the boundary layer with a specified thickness $\tau$:

$$\theta(d, \tau) = \begin{cases} \frac{d - (1-\tau)}{d_{\max} - (1-\tau)} & \text{if } d > 1 - \tau \\ 0 & \text{otherwise} \end{cases} \tag{16}$$

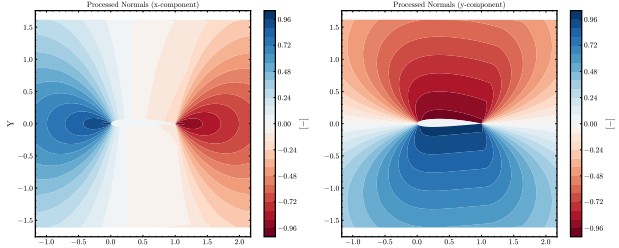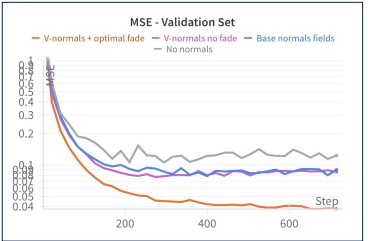

(a) Visualization of the volumetric normals field around an airfoil, showing the x-component (left) and y-component (right) with a smooth transition from the surface to the far field.

(b) Validation loss comparison when volumetric normals are incorporated as input features.

Figure 17: Volumetric normals field and its impact on model performance. The processed normals provide directional information that helps the network better understand the relationship between each point in the domain and the airfoil geometry.

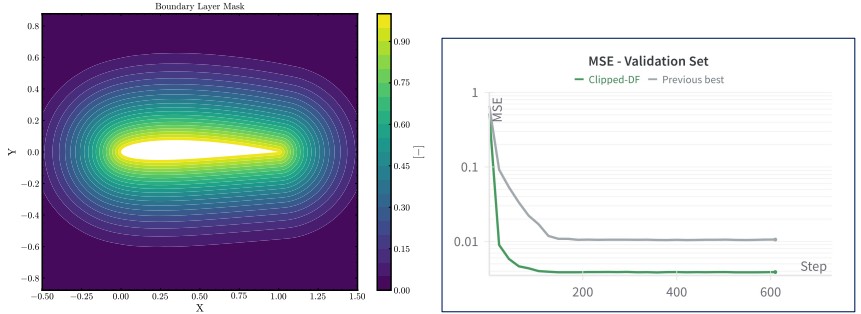

(a) Visualization of the boundary layer mask. The mask uses a squared decay function to create a smooth transition.

(b) Training loss comparison showing the improvement in model performance when the boundary layer mask is incorporated.

Figure 18: Boundary layer mask and its impact on model performance. The mask helps focus the model's attention on the critical near-wall region where flow variables exhibit the steepest gradients.

3. We apply a decay function to create a smooth transition from the surface. For squared decay:

$$\mathrm{BL}_{\mathrm{mask}} = \theta(\hat{d}, \tau)^2 \tag{17}$$

Figure 18a shows the resulting boundary layer mask, highlighting the near-wall region with varying intensity based on distance from the surface, and the reduced training loss compared to the baseline model.

More in detail, the effect of the boundary layer mask on MARIO's predictive performance, is evaluated on the scarce task. As shown in Table 7, removing the boundary layer mask causes drag coefficient errors to increase nearly sixfold, while lift prediction remains relatively robust. This asymmetric impact reflects the physics of airfoil aerodynamics: drag coefficients depend heavily on accurate boundary layer resolution, while lift forces are predominantly driven by pressure distribution governed by potential flow effects, especially at moderate angles of attack. Figure 19 illustrates

| Model | $C_D$ | $C_L$ | $\rho_D$ | $\rho_L$ |
|---|---|---|---|---|
| MARIO with BL mask | **0.794** | **0.115** | **0.102** | **0.997** |
| MARIO without BL mask | 4.780 | 0.151 | -0.074 | 0.995 |

Table 7: **Effect of boundary layer mask on aerodynamic coefficient predictions.** $C_D$ and $C_L$ values represent Mean Relative Error (%), while $\rho_D$ and $\rho_L$ are Spearman's rank correlation coefficients. Bold values indicate best performance.

how the boundary layer mask affects flow prediction quality across the boundary layer at mid-chord

position. Without the mask, the model exhibits noticeable oscillations in velocity and turbulent viscosity profiles. These artifacts arise because the network must use a single set of weights to simultaneously capture both steep near-wall gradients and smoother outer flow behavior. By contrast, the model with boundary layer mask accurately resolves the sharp velocity gradient and turbulent viscosity peak without introducing spurious oscillations. The boundary layer mask effectively allows the network to adapt its local sensitivity according to spatial position—maintaining high responsiveness near the wall where gradients are steep, while enforcing smoother behavior in the outer flow. This feature engineering approach significantly enhances the model's ability to capture the essential characteristics of boundary layer flows without requiring excessive model capacity or resolution.

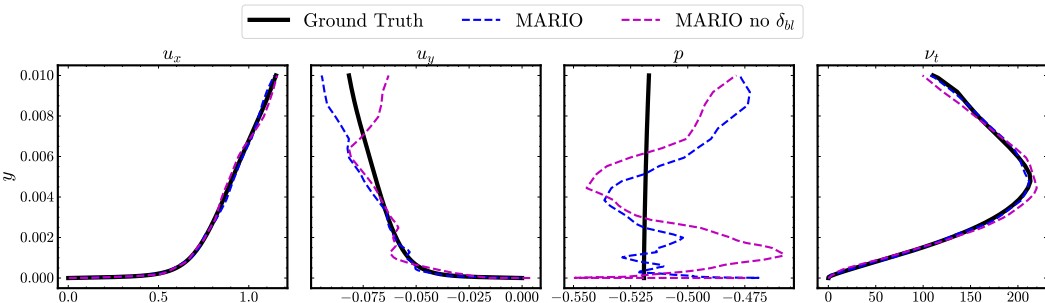

Figure 19: Comparison of velocity, pressure, and turbulent viscosity profiles across the boundary layer region at x=0.5c, contrasting predictions with and without the boundary layer mask. The model with boundary layer mask (blue) more accurately captures the steep gradients near the wall without introducing oscillations seen in the model without the mask (orange). Flow conditions: $\alpha = 7.91°$, $M = 0.16$.

Some illustrations of the predicted fields are provided in Figures 20-21.

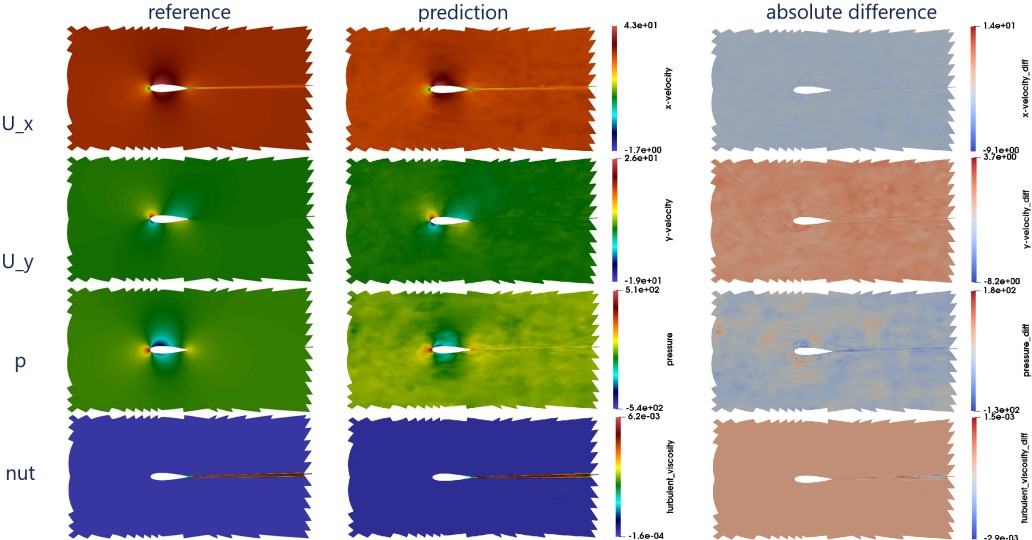

Figure 20: Illustration of the result on the first sample of the test set for the MARIO approach.

## Appendix I   GeoMPNN Details

Input node features $z_{\text{in}}(x)$ for mesh point $x$ include the inlet velocity $v_\infty$, orthogonal distance from the airfoil $d(x)$, surface normals $n(x)$, coordinates $x$ and distance $\|x\|$ relative the leading edge of

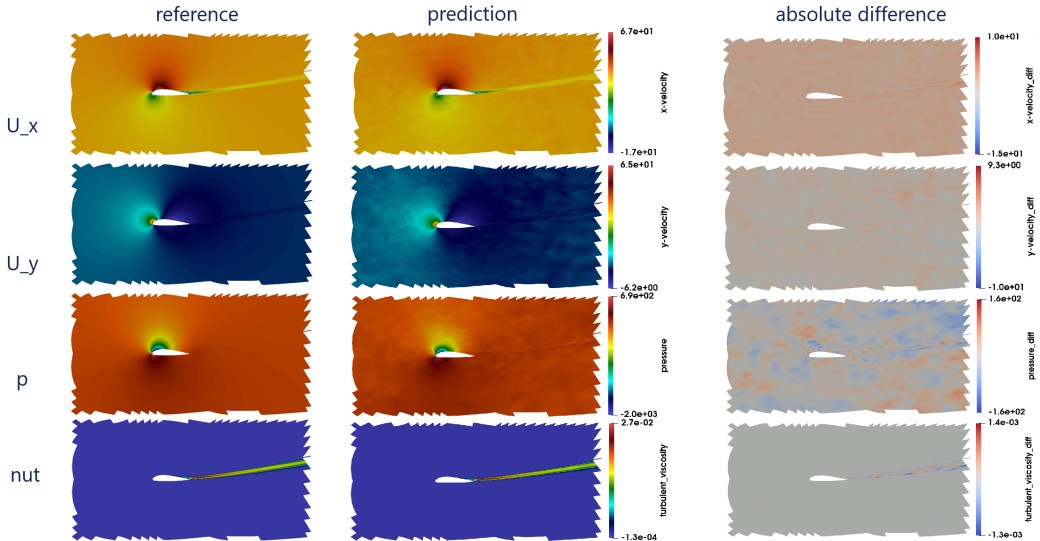

Figure 21: Illustration of the result on the first sample of the test_ood set for the MARIO approach.

the airfoil as

$$z_{\text{in}}(x) = [z_{\text{base}}(x), z_{\text{trail}}(x), z_{\text{sph}}(x), z_{\text{sine}}(x)] \in \mathbb{R}^{251} \tag{18}$$

$$z_{\text{base}}(x) := [x, n(x), v_\infty, d(x), \|x\|] \in \mathbb{R}^8. \tag{19}$$

Input edge features $e_{\text{in}}(y, x)$ for the directed edge from $y$ to $x$ include displacement $y - x$ and distance $\|y - x\|$ as

$$e_{\text{in}}(y, x) = [e_{\text{base}}(y, x), e_{\text{sph}}(y, x), e_{\text{sine}}(y, x)] \in \mathbb{R}^{115} \tag{20}$$

$$e_{\text{base}}(y, x) := [y - x, \|y - x\|] \in \mathbb{R}^3.$$

In the following sections, we detail the remaining components $z_{\text{trail}}(x)$, $z_{\text{sph}}(x)$ and $z_{\text{sine}}(x)$ of the input node features in Equation (18), as well as $e_{\text{sph}}(y, x)$ and $e_{\text{sine}}(y, x)$ from the input edge features in Equation (20).

## I.1 Augmented Coordinate System

In this section, we describe enhancements to our coordinate system representation to more faithfully represent relations between the airfoil geometry and mesh regions in which interactions with the airfoil give rise to distinct patterns of dynamics. In Section I.1.1, we introduce a trailing edge coordinate system, while in Section I.1.2, we discuss the addition of angles to input features.

### I.1.1 Trailing Edge Coordinate System

The mesh coordinates $\mathcal{X} \subset \mathbb{R}^2$ are constructed such that the origin is the leading edge $x_{\text{lead}}$ of the airfoil, that is, the leftmost point on the airfoil surface mesh $\mathcal{X}_{\text{surf}} \subset \mathcal{X}$, or more formally

$$x_{\text{lead}} := \arg\min_{x \in \mathcal{X}_{\text{surf}}} x = 0,$$

where $x$ denotes the first component of $x = [x \quad y]^\top$. We can similarly define the trailing edge of the airfoil $x_{\text{trail}}$ as the rightmost point on $\mathcal{X}_{\text{surf}}$ by

$$x_{\text{trail}} := \arg\max_{x \in \mathcal{X}_{\text{surf}}} x.$$

The mesh $\mathcal{X}$ can now be partitioned along the $x$-axis into three subsets of points as

$$\mathcal{X} = \mathcal{X}_{\text{fs}} \cup \mathcal{X}_{\text{af}} \cup \mathcal{X}_{\text{ds}},$$

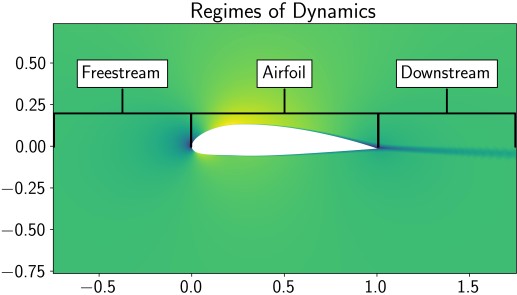

Figure 22: Spatial regimes of dynamics. Interactions between the airfoil and flow are unique in each of these three regions, creating distinct patterns of dynamics. Therefore, coordinate system representations should enable the model to distinguish between each of these regions.

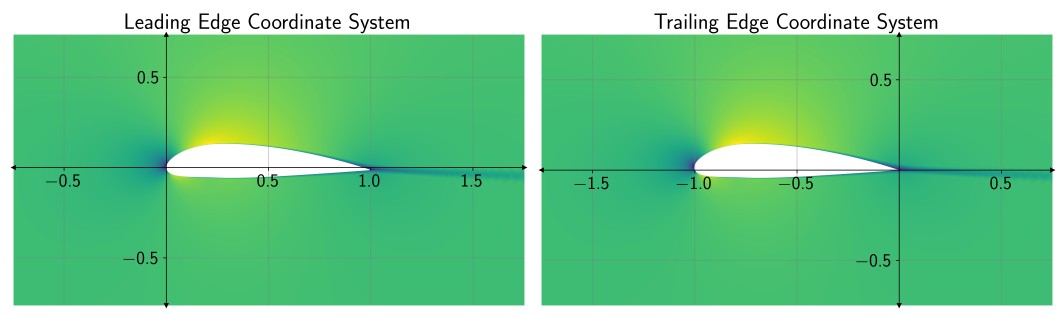

(a) Leading edge coordinate system.  (b) Trailing edge coordinate system.

Figure 23: Leading and trailing edge coordinate systems. The leading and trailing edge coordinate systems enable the model to be able to distinguish between the three regimes of dynamics described in Figure 22.

each of which are defined as

$$\mathcal{X}_{\text{fs}} \coloneqq \{\boldsymbol{x} : \boldsymbol{x} \in \mathcal{X}, x < x_{\text{lead}}\} \qquad \mathcal{X}_{\text{af}} \coloneqq \{\boldsymbol{x} : \boldsymbol{x} \in \mathcal{X}, x \in [x_{\text{lead}}, x_{\text{trail}}]\}$$

$$\mathcal{X}_{\text{ds}} \coloneqq \{\boldsymbol{x} : \boldsymbol{x} \in \mathcal{X}, x > x_{\text{trail}}\}.$$

As can be seen in Figure 22, the dynamics occurring in each of these regions are distinct from one another. In the freestream region $\mathcal{X}_{\text{fs}}$, the behavior of the flow is largely independent of the airfoil. Upon entering $\mathcal{X}_{\text{af}}$, the airfoil surface deflects, compresses, and slows down air particles. These interactions are then carried downstream of the airfoil in the region described by $\mathcal{X}_{\text{ds}}$, where particles are still influenced by the airfoil geometry but no longer come into direct contact with its surface. It is therefore important for the model to be able to distinguish which partition a given mesh point belongs to for faithful modeling of the dynamics.

Given the base node features $\boldsymbol{z}_{\text{base}}(\boldsymbol{x})$ for the mesh point $\boldsymbol{x} = \begin{bmatrix} x & y \end{bmatrix}^{\top}$ shown in Equation (19), it is straightforward for the model to determine whether $\boldsymbol{x} \in \mathcal{X}_{\text{fs}}$ or $\boldsymbol{x} \in \mathcal{X}_{\text{af}} \cup \mathcal{X}_{\text{ds}}$ via $\text{sign}(x)$, as can be seen in Figure 23a. However, distinguishing between $\mathcal{X}_{\text{af}}$ and $\mathcal{X}_{\text{ds}}$ is more challenging, as unlike $x_{\text{lead}}$, $x_{\text{trail}}$ can vary across airfoil geometries, for example according to the length $x_{\text{trail}} - x_{\text{lead}}$ of the airfoil. We therefore introduce a trailing edge coordinate system $\mathcal{X}_{\text{trail}} \subset \mathbb{R}^2$ to better enable the model to distinguish between these regions. The trailing edge coordinate system, shown in Figure 23b, places the origin at $\boldsymbol{x}_{\text{trail}} \in \mathcal{X}$, and is defined as

$$\mathcal{X}_{\text{trail}} \coloneqq \{\boldsymbol{x} - \boldsymbol{x}_{\text{trail}} : \boldsymbol{x} \in \mathcal{X}\}.$$

We include the trailing edge coordinate system in the input node features as

$$\boldsymbol{z}_{\text{trail}}(\boldsymbol{x}) \coloneqq [\boldsymbol{x}_{\tau}, \|\boldsymbol{x}_{\tau}\|] \in \mathbb{R}^3,$$

where $\boldsymbol{x}_{\tau} \coloneqq \boldsymbol{x} - \boldsymbol{x}_{\text{trail}} \in \mathcal{X}_{\text{trail}}$.

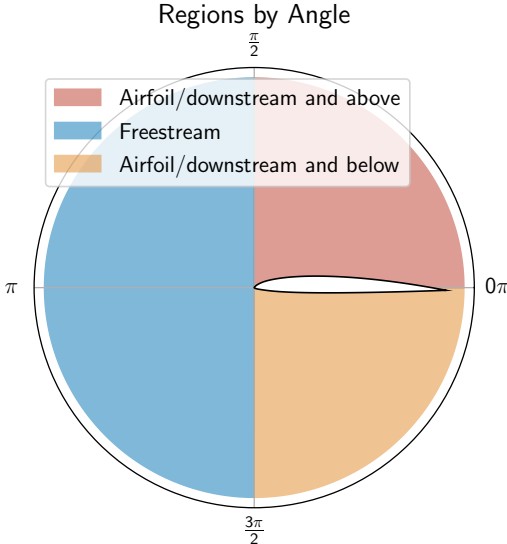

Figure 24: Regions by angle in the leading edge coordinate system. Polar angles can be used to derive both which of the regions along the $x$-axis defined in Figure 22 a mesh point is in, as well as where the point is vertically with respect to the airfoil. The same information is contained in the Cartesian form of the leading edge coordinate system, although the model must learn interactions between the $x$ and $y$ coordinates in order to derive it.

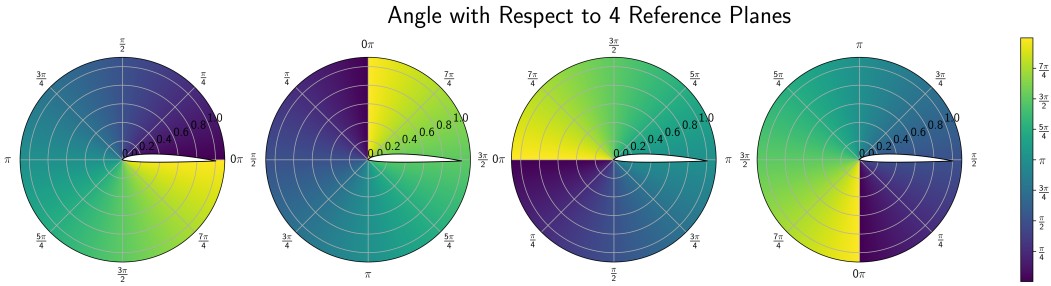

Figure 25: Angles in the leading edge coordinate system with respect to four different reference planes. Features in the POLAR model include angles with respect to all four reference planes.

### I.1.2 Hybrid Polar-Cartesian Coordinates

Although the trailing and leading edge coordinate systems enable GeoMPNN to distinguish between the regions defined in Figure 22, other distinctive regions are present which require the model to derive interactions between horizontal and vertical coordinates. For example, for $\boldsymbol{x} = \begin{bmatrix} x & y \end{bmatrix}^\top \in \mathcal{X}$, $\mathrm{sign}(x)$ determines whether the mesh point is in $\mathcal{X}_{\mathrm{fs}}$ or $\mathcal{X}_{\mathrm{af}} \cup \mathcal{X}_{\mathrm{ds}}$, while $\mathrm{sign}(y)$ indicates whether the mesh point is above or below the surface of the airfoil. If $\boldsymbol{x} \in \mathcal{X}_{\mathrm{fs}}$, then whether it is above or below the airfoil largely has no effect on the resultant dynamics, although if $\boldsymbol{x} \in \mathcal{X}_{\mathrm{af}} \cup \mathcal{X}_{\mathrm{ds}}$, the dynamics above the airfoil are unique from those below, *e.g.,* the generation of lift requires pressure on the lower surface of the airfoil to be greater than pressure on the upper surface [47].

To simplify learning, we extend model inputs to include the polar angle of $\boldsymbol{x} = \begin{bmatrix} x & y \end{bmatrix}^\top$ with respect to polar axis $\begin{bmatrix} 1 & 0 \end{bmatrix}^\top$ as $\theta(\boldsymbol{x}) \coloneqq \mathrm{atan2}(y, x)$. As $\boldsymbol{z}_{\mathrm{in}}(\boldsymbol{x})$ also includes $\|\boldsymbol{x}\|$, the inputs can be seen to now contain both polar and Cartesian coordinates. Instead of learning interactions between coordinates, this enables derivation of relationships directly from $\theta(\boldsymbol{x})$. As shown in Figure 24, this simplification can be seen through the previous example, where $\theta(\boldsymbol{x}) \in (0, \frac{\pi}{2}]$ or $(\frac{3\pi}{2}, 2\pi]$ indicates that $\boldsymbol{x}$ is in $\mathcal{X}_{\mathrm{af}} \cup \mathcal{X}_{\mathrm{ds}}$ and above or below the airfoil, respectively, while $\theta(\boldsymbol{x}) \in (\frac{\pi}{2}, \frac{3\pi}{2}]$ indicates that $\boldsymbol{x}$ is in $\mathcal{X}_{\mathrm{fs}}$.

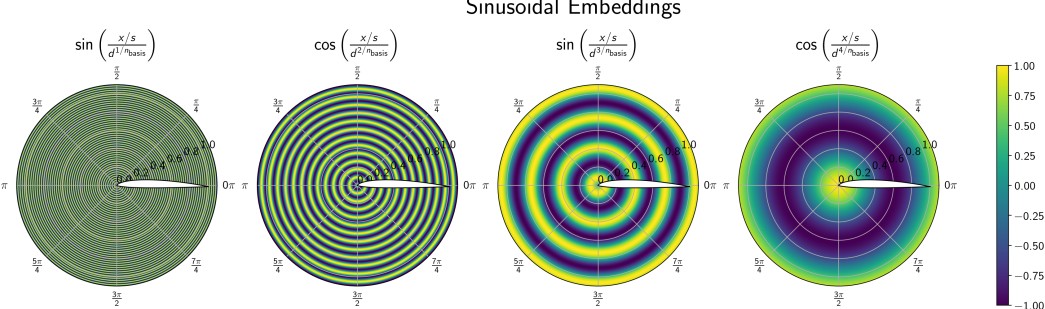

Figure 26: Sinusoidal embeddings of varying frequency for distance in the leading edge coordinate system.

Instead of only using angle with respect to one reference plane, we compute angles with respect to 4 different polar axes $[1 \quad 0]^\top, [0 \quad 1]^\top, [-1 \quad 0]^\top$ and $[0 \quad -1]^\top$ as shown in Figure 25. Angles are then computed as

$$\theta^4(\boldsymbol{x}) := [\theta(\boldsymbol{x}), \theta(\boldsymbol{R}_{90^\circ}\boldsymbol{x}), \theta(\boldsymbol{R}_{180^\circ}\boldsymbol{x}), \theta(\boldsymbol{R}_{270^\circ}\boldsymbol{x})] \in \mathbb{R}^4,$$

where $\boldsymbol{R}_{\nu^\circ}$ denotes a rotation by $\nu$ degrees. We include polar angles in the input node features as

$$\boldsymbol{z}_{\text{polar}}(\boldsymbol{x}) := \left[\theta^4(\boldsymbol{x}), \theta^4(\boldsymbol{x}_\tau)\right] \in \mathbb{R}^8.$$

Furthermore, we also add the angle of edges with respect to the polar axes to input edge features as

$$\boldsymbol{e}_{\text{polar}}(\boldsymbol{y}, \boldsymbol{x}) := \theta^4(\boldsymbol{y} - \boldsymbol{x}) \in \mathbb{R}^4.$$

Note that neither $\boldsymbol{z}_{\text{polar}}(\boldsymbol{x})$ nor $\boldsymbol{e}_{\text{polar}}(\boldsymbol{y}, \boldsymbol{x})$ appear in Equations (18) and (20), as they are nested inside $\boldsymbol{z}_{\text{sph}}(\boldsymbol{x})$ and $\boldsymbol{e}_{\text{sph}}(\boldsymbol{y}, \boldsymbol{x})$, respectively, which are introduced in Section I.1.5.

### I.1.3 Coordinate System Embedding

To enhance the expressiveness of input coordinates $x \in \mathbb{R}$, we expand them to $[B_1(x), B_2(x), \ldots, B_{n_{\text{basis}}}(x)]$, where $\{B_i\}_{i=1}^{n_{\text{basis}}}$ is a $n_{\text{basis}}$-dimensional basis. In the first linear layer $\boldsymbol{W}$ of the embedding module, the model can then learn arbitrary functions of the coordinates $\sum_{i=1}^{n_{\text{basis}}} w_i B_i(x)$. In Section I.1.4, we describe our embedding of Cartesian coordinates and distances using sinusoidal basis functions, while in Section I.1.5, we discuss spherical harmonic angle embeddings. In both cases, we take $n_{\text{basis}} = 8$.

### I.1.4 Sinusoidal Basis

For Cartesian coordinates and distances $x$, we employ the sinusoidal basis commonly used in Transformer models [48] given by

$$\text{PE}(x) := \left[\sin\left(\frac{x/s}{d^{i/n_{\text{basis}}}}\right), \cos\left(\frac{x/s}{d^{i/n_{\text{basis}}}}\right)\right]_{i=0}^{n_{\text{basis}}-1} \in \mathbb{R}^{2n_{\text{basis}}}. \tag{21}$$

The division by $s$ in the numerator of Equation (21) is to account for the spacing of the grid points, as [48] developed this scheme for $x \in \mathbb{Z}^+$ as a sequence position. While the spacing between consecutive sequence positions is 1 for the setting considered by [48], for a computational mesh, the spacing between adjacent points can be substantially smaller. We furthermore set $d$ dependent on $s$ and the size of the domain $L$ as $d = \frac{4L}{s\pi}$. Sinusoidal embeddings for the distance in the leading edge coordinate system are shown in Figure 26.

We add sinusoidal embeddings of coordinates and distances to input node features as

$$\boldsymbol{z}_{\text{sine}}(\boldsymbol{x}) := [\text{PE}(\boldsymbol{x}), \text{PE}(\boldsymbol{x}_\tau), \text{PE}(d(\boldsymbol{x})), \text{PE}(\|\boldsymbol{x}\|), \text{PE}(\|\boldsymbol{x}_\tau\|)] \in \mathbb{R}^{112},$$

where with a slight abuse of notation, we vectorize PE as

$$\text{PE}(\boldsymbol{x}) := [\text{PE}(x), \text{PE}(y)] \in \mathbb{R}^{4n_{\text{basis}}}. \tag{22}$$

Edge displacements and distances are similarly embedded as

$$\boldsymbol{e}_{\text{sine}}(\boldsymbol{y}, \boldsymbol{x}) := [\text{PE}(\|\boldsymbol{y} - \boldsymbol{x}\|), \text{PE}(\boldsymbol{y} - \boldsymbol{x})] \in \mathbb{R}^{48}.$$

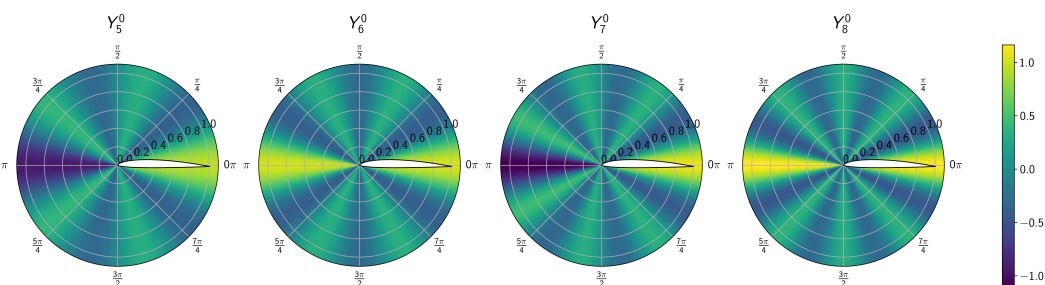

Figure 27: Spherical harmonics embeddings of angles in the leading edge coordinate system with $m = 0$ and varying order $\ell$.

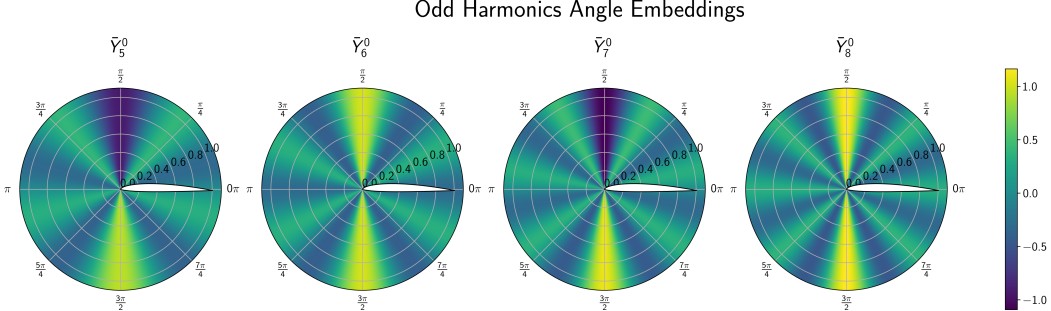

Figure 28: Odd harmonics embeddings of angles in the leading edge coordinate system with $m = 0$ and varying order $\ell$.

### I.1.5 Spherical Harmonics Basis

For angles $\theta$, we instead use $m = 0$ spherical harmonics embeddings [49, 50]. The $m = 0$ order $\ell$ spherical harmonic is given by [51]

$$Y_\ell^0(\theta) := \sqrt{\frac{(2\ell + 1)!}{4\pi}} P_\ell(\cos(\theta)) = \sum_{k=1}^{\ell} C_{\ell,k} \cos^k(\theta), \tag{23}$$

where the coefficients $C_{\ell,k} \in \mathbb{R}$ are obtained from the order $\ell$ Legendre polynomial $P_\ell$. However, as can be seen in Equation (23), since $Y_\ell^0(\theta)$ is composed of powers of cosine, it is an even function, and as such, it cannot distinguish between $\theta$ and $-\theta$. We therefore define a set of odd basis functions $\bar{Y}_\ell^0(\theta)$ as

$$\bar{Y}_\ell^0(\theta) := \sqrt{\frac{(2\ell + 1)!}{4\pi}} P_\ell(\sin(\theta)). \tag{24}$$

As can be seen in Figure 28, the odd harmonics can distinguish between $-\theta$ and $\theta$. We obtain embeddings of $\theta$ using both sets of bases in Equations (23) and (24) as

$$\mathrm{SpH}(\theta) := \left[ Y_i^0(\theta), \bar{Y}_i^0(\theta) \right]_{i=1}^{n_{\mathrm{basis}}} \in \mathbb{R}^{2n_{\mathrm{basis}}}.$$

We embed the angle features $\boldsymbol{z}_{\mathrm{polar}}(\boldsymbol{x})$ introduced in Section I.1.2 in these bases as

$$\boldsymbol{z}_{\mathrm{sph}}(\boldsymbol{x}) := \left[ \mathrm{SpH}(\theta^4(\boldsymbol{x})), \mathrm{SpH}(\theta^4(\boldsymbol{x}_\tau)) \right] \in \mathbb{R}^{128},$$

where SpH is vectorized as in Equation (22) as

$$\mathrm{SpH}(\theta^4(\boldsymbol{x})) := [\mathrm{SpH}(\theta(\boldsymbol{x})), \mathrm{SpH}(\theta(\boldsymbol{R}_{90°}\boldsymbol{x})), \mathrm{SpH}(\theta(\boldsymbol{R}_{180°}\boldsymbol{x})), \mathrm{SpH}(\theta(\boldsymbol{R}_{270°}\boldsymbol{x}))] \in \mathbb{R}^{8n_{\mathrm{basis}}}.$$

Edge angles in $\boldsymbol{e}_{\mathrm{polar}}(\boldsymbol{y}, \boldsymbol{x})$ are also embedded in these bases as

$$\boldsymbol{e}_{\mathrm{sph}}(\boldsymbol{y}, \boldsymbol{x}) := \mathrm{SpH}(\theta^4(\boldsymbol{y} - \boldsymbol{x})) \in \mathbb{R}^{64}.$$

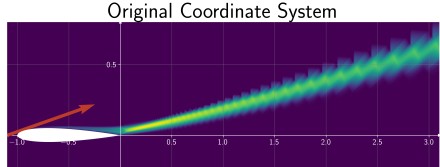 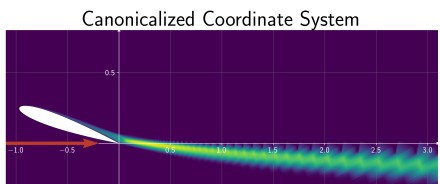 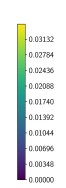

Figure 29: Turbulent viscosity and inlet velocity in the trailing edge coordinate system (*left*) and in the canonicalized trailing edge coordinate system (*right*). The direction of the inlet velocity, shown in red, is strongly associated with the direction of the non-zero parts of the turbulent viscosity field. After canonicalizing the coordinate system such that the inlet velocity is parallel with the $x$-axis, the non-zero parts of the turbulent viscosity are concentrated in the region around the $x$-axis for all examples.

## I.2 Turbulent Viscosity and Pressure Transformations

In the previous sections, we have discussed techniques that we apply to all four GeoMPNN models trained to predict velocities $\bar{u}_x$ and $\bar{u}_y$, pressure $\bar{p}$, and turbulent viscosity $\nu_t$. However, we found the pressure and turbulent viscosity fields to be particularly challenging to accurately model compared to the velocity fields. We therefore introduce specific techniques for enhancing model generalization on each of these fields. In Section I.2.1, we derive a change of basis to canonicalize input features with respect to inlet velocity. We additionally re-parameterize our model to predict the log-transformed pressure, as discussed in Section I.2.2.

### I.2.1 Inlet Velocity Canonicalization

As can be seen in Figure 29, the non-zero regions of the turbulent viscosity $\nu_t$ are strongly associated with the direction of the inlet velocity $\boldsymbol{v}_\infty$. To improve generalization on this field, we apply a change-of-basis to obtain a new coordinate system under which $\boldsymbol{v}_\infty$ is rotated to be parallel with the $x$-axis. This effectively *canonicalizes* the coordinate system with respect to the inlet velocity, a technique which has been applied in machine learning to improve generalization [52, 53]. In the case of $\nu_t$, as can be seen in Figure 29, the association between $\boldsymbol{v}_\infty$ and $\nu_t$ implies that many of the non-zero parts of $\nu_t$ will occur in the region along the $x$-axis in the canonicalized coordinate system, resulting in a substantially less difficult modeling task.

This canonicalization can be achieved with the rotation matrix $\boldsymbol{R_v} \in O(2)$ given by

$$\boldsymbol{R_v} \coloneqq \frac{1}{\|\boldsymbol{v}_\infty\|} \begin{bmatrix} v_1 & v_2 \\ -v_2 & v_1 \end{bmatrix},$$

where $\boldsymbol{v}_\infty = \begin{bmatrix} v_1 & v_2 \end{bmatrix}^\top$. $\boldsymbol{R_v}$ is obtained by deriving the matrix that rotates $\boldsymbol{v}_\infty$ to be parallel with the $x$-axis, that is

$$\boldsymbol{R_v}\boldsymbol{v}_\infty = [\|\boldsymbol{v}_\infty\| \quad 0]^\top .$$

For the GeoMPNN models trained to predict turbulent viscosity and pressure, we extend the input node features $\boldsymbol{z}_{\text{in}}(\boldsymbol{x})$ to additionally include coordinates, angles, and surface normals in the canonicalized coordinate system as

$$\boldsymbol{z}_{\text{inlet}}(\boldsymbol{x}) \coloneqq \left[ \boldsymbol{R_v}\boldsymbol{x}, \boldsymbol{R_v}\boldsymbol{x}_\tau, \text{PE}(\boldsymbol{R_v}\boldsymbol{x}), \text{PE}(\boldsymbol{R_v}\boldsymbol{x}_\tau), \boldsymbol{R_v}\boldsymbol{n}(\boldsymbol{x}), \text{SpH}(\theta^4(\boldsymbol{R_v}\boldsymbol{x})), \text{SpH}(\theta^4(\boldsymbol{R_v}\boldsymbol{x}_\tau)) \right] ,$$

with $\boldsymbol{z}_{\text{inlet}}(\boldsymbol{x}) \in \mathbb{R}^{198}$. We additionally add the canonicalized edge displacements and angles to the input edge features $\boldsymbol{e}_{\text{in}}(\boldsymbol{y}, \boldsymbol{x})$ as

$$\boldsymbol{e}_{\text{inlet}}(\boldsymbol{y}, \boldsymbol{x}) \coloneqq \left[ \boldsymbol{R_v}(\boldsymbol{y} - \boldsymbol{x}), \text{PE}(\boldsymbol{R_v}(\boldsymbol{y} - \boldsymbol{x})), \text{SpH}(\theta^4(\boldsymbol{R_v}(\boldsymbol{y} - \boldsymbol{x}))) \right] \in \mathbb{R}^{98}.$$

### I.2.2 Log-Transformed Pressure Prediction

Lift forces acting on the airfoil are primarily generated through the difference between pressure on the lower and upper surfaces of the airfoil [47]. Therefore, accurate modeling of the pressure $\bar{p}$ is necessary for accurate prediction of the lift coefficient $C_L$. However, this large differential leads to a more challenging prediction target. As can be seen in Figures 30 and 31a, there is a large amount of

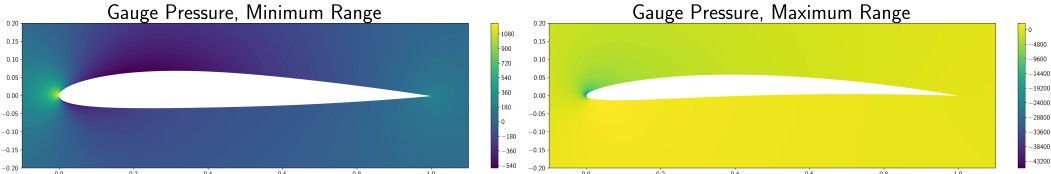

Figure 30: Pressure field with the least range (*left*) and with the greatest range (*right*).

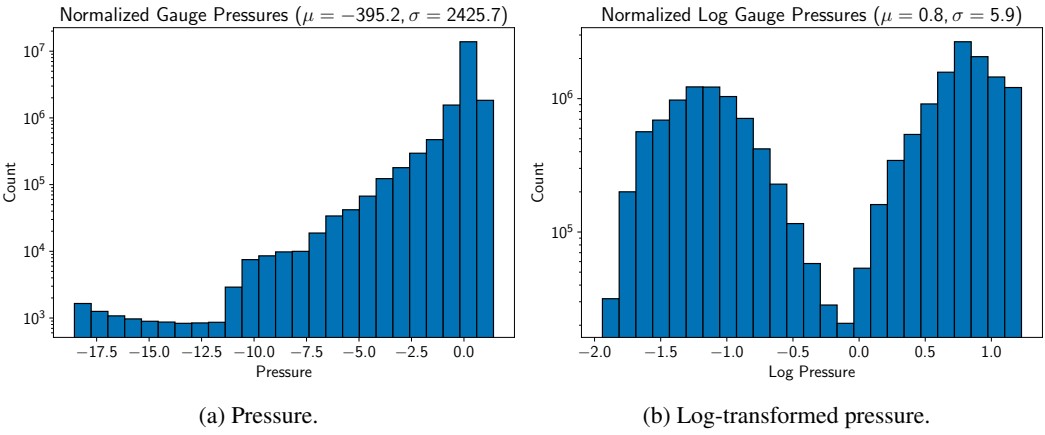

(a) Pressure.            (b) Log-transformed pressure.

Figure 31: Distribution of normalized pressures.

variance between the low pressure region above the airfoil and the high pressure region below the airfoil. Note that $\bar{p}(\boldsymbol{x})$ is the *gauge pressure*, which represents the deviation of the absolute pressure $p_{\text{abs}}$ from the freestream pressure $p_\infty$ as

$$\bar{p}(\boldsymbol{x}) := p_{\text{abs}}(\boldsymbol{x}) - p_\infty. \tag{25}$$

While log transformations are often used to reduce the variance in positive-valued data, the gauge pressure can also take negative values, and thus, we introduce a log-transformation of the pressure as

$$\bar{q}(\boldsymbol{x}) := \text{sign}(\bar{p}(\boldsymbol{x})) \log(|\bar{p}(\boldsymbol{x})| + 1).$$

As can be seen in Figures 31b and 32, this substantially reduces variance. We therefore train GeoMPNN to predict the log-transformed pressure field $\bar{q}(\boldsymbol{x})$. The untransformed pressure $\bar{p}$ can then be obtained from $\bar{q}$ by applying the inverse transformation given by

$$\bar{p}(\boldsymbol{x}) = \text{sign}(\bar{q}(\boldsymbol{x}))(\exp(|\bar{q}(\boldsymbol{x})|) - 1).$$

Some illustrations of the predicted fields are provided in Figures 33-34.

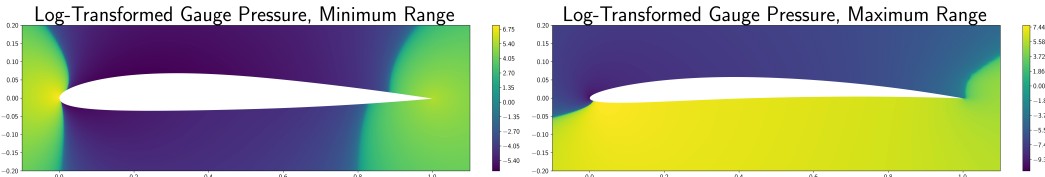

Figure 32: Log-transformed pressure field with the least range (*left*) and with the greatest range (*right*).

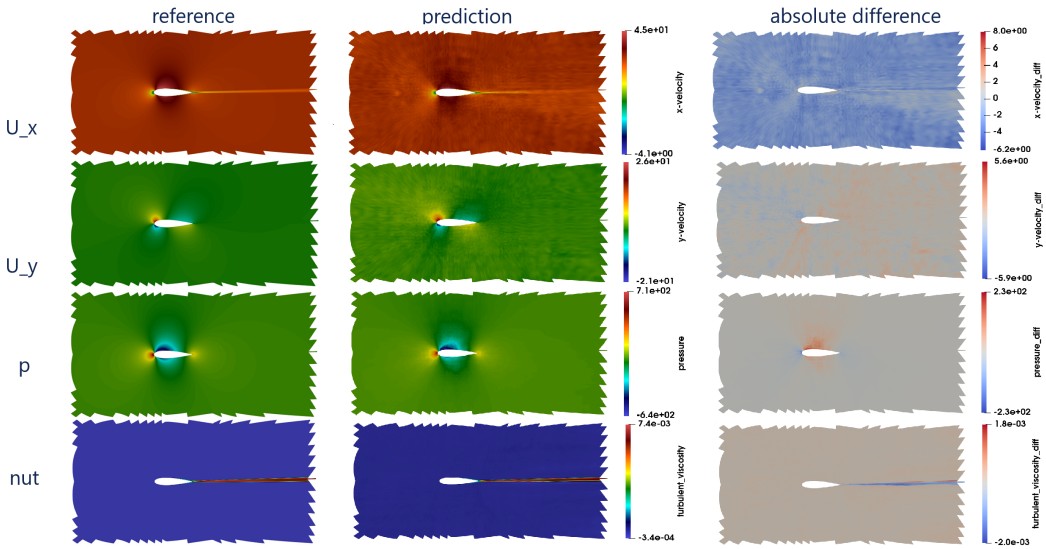

Figure 33: Illustration of the result on the first sample of the test set for the GeoMPNN approach.

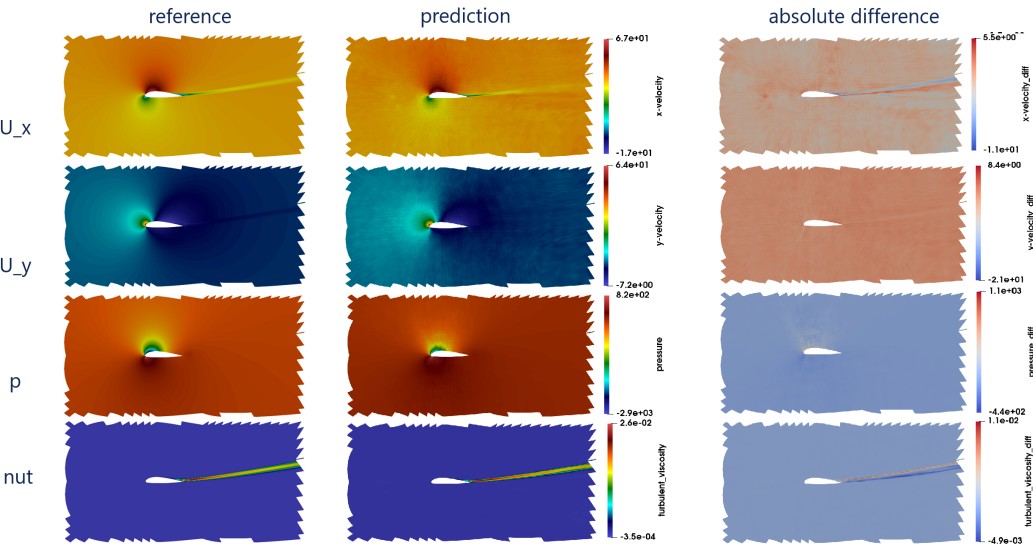

Figure 34: Illustration of the result on the first sample of the test_ood set for the GeoMPNN approach.

# NeurIPS Paper Checklist

The checklist is designed to encourage best practices for responsible machine learning research, addressing issues of reproducibility, transparency, research ethics, and societal impact. Do not remove the checklist: **The papers not including the checklist will be desk rejected.** The checklist should follow the references and follow the (optional) supplemental material. The checklist does NOT count towards the page limit.

Please read the checklist guidelines carefully for information on how to answer these questions. For each question in the checklist:

- You should answer [Yes] , [No] , or [NA] .
- [NA]  means either that the question is Not Applicable for that particular paper or the relevant information is Not Available.
- Please provide a short (1–2 sentence) justification right after your answer (even for NA).

**The checklist answers are an integral part of your paper submission.** They are visible to the reviewers, area chairs, senior area chairs, and ethics reviewers. You will be asked to also include it (after eventual revisions) with the final version of your paper, and its final version will be published with the paper.

The reviewers of your paper will be asked to use the checklist as one of the factors in their evaluation. While "[Yes] " is generally preferable to "[No] ", it is perfectly acceptable to answer "[No] " provided a proper justification is given (e.g., "error bars are not reported because it would be too computationally expensive" or "we were unable to find the license for the dataset we used"). In general, answering "[No] " or "[NA] " is not grounds for rejection. While the questions are phrased in a binary way, we acknowledge that the true answer is often more nuanced, so please just use your best judgment and write a justification to elaborate. All supporting evidence can appear either in the main paper or the supplemental material, provided in appendix. If you answer [Yes]  to a question, in the justification please point to the section(s) where related material for the question can be found.

IMPORTANT, please:

- **Delete this instruction block, but keep the section heading "NeurIPS paper checklist",**
- **Keep the checklist subsection headings, questions/answers and guidelines below.**
- **Do not modify the questions and only use the provided macros for your answers**.

