# OpenReview forum: "ML4CFD Competition: Results and Retrospective Analysis"
_NeurIPS.cc/2025/Datasets_and_Benchmarks_Track — NeurIPS 2025 Datasets and Benchmarks Track poster_

### Official Review · Reviewer_pKDA · 2025-06-11

**Rating:** 5
**Confidence:** 5

**Summary:**

This is a retrospective paper on the 2024 NeurIPS ML4CFD competition, which lasted from July 1st 2024 to October 26th 2024. 240 teams made 650 submissions to the competition. The results were presented at the 2024 Neurips conference (see https://neurips.cc/virtual/2024/competition/84799) and are summarized and discussed in this paper for the 2025 conference.

The competition is based on the airfRANS (ha ha very funny) dataset, which gives the steady-state subsonic flow profiles of the 2D incompressible Navier-stokes equations over NACA airfoils. This is a classic problem in computational fluid dynamics which traditional numerical methods can accurately solve. This dataset was introduced at the 2022 NeurIPS datasets and benchmarks track. The goal is to accurately reproduce the flow fields (velocity, pressure, turbulent viscosity) outside the airfoil, as the Reynolds number (Re) and angle of attack (AoA) are varied.

The competition results are scored using the Learning Industrial Physical Simulation (LIPS) benchmarking framework, which uses a weighted average of multiple metrics to output a score between 0% and 100%. This framework was also introduced at the 2022 NeurIPS datasets and benchmarks track. The metrics are in-distribution accuracy, out-of-distribution accuracy, so-called "physics compliance", and relative speedup compared to the OpenFOAM numerical method baseline.

The four top-scoring submissions from the competition are discussed in the paper. The paper writes that "the top entry exceeded the performance of the original OpenFOAM solver on aggregate metrics." The original OpenFOAM solver scored 82.5%, while the top entry scored 84.7%.

This competition is in many ways an extremely strong contribution to the field. However, it is also deeply flawed in important ways. Unfortunately, it is far too late to fix the flaws of the competition. However, it is not too late to fix the flaws of this paper, by making clear to the reader what the flaws of the competition are. I will give the authors suggestions for what should be changed.

I also have many questions about the competition which are not answered by the paper, the references, the website, or the accompanying Jupyter notebooks. The paper could thus do a better job explaining certain aspects of the competition.

In my judgement, this paper should be accepted based on the strengths of the competition. I thus gave the paper a score of a 5. Nevertheless, I hope the authors will take my suggestions for improving the paper and for more openly and honestly discussing the significant flaws in what is otherwise a great competition. If that happens, then not only will this have been a great competition, but this will become a great paper.

While I judge that this paper should be accepted, I believe there is also a good argument to be made that accepting this paper to NeurIPS doesn't add much value to the community, and thus should be rejected. I will present that argument in the "Additional Feedback" section. If the AC recommends rejecting the paper based on that argument, I am comfortable doing so.

**Additional Feedback:**

# An Argument for Rejecting This Paper

While in my judgement this paper should be accepted, I believe there is a good argument to be made that this paper should be rejected.

The reason is that the *results from this competition have already been presented at NeurIPS 2024*. See https://neurips.cc/virtual/2024/competition/84799. If the results from this competition were presented for 3 hours (1:30pm-4:30pm) at NeurIPS 2024, then what value to the community does this paper add if the exact same results are presented again at NeurIPS 2025? With limited spots at NeurIPS, perhaps the community would benefit more from accepting a different paper?

**Dataset Code Accessibility:**

Yes

**Dataset Code Comments:**

I read through the website, looked at the Jupyter notebooks, and thought they did a great job with the code, documentation, explanations, etc.

**Ethical Considerations:**

No, there are no or only very minor ethics concerns

**Final Justification:**

This paper is a retrospective on a ML4CFD competition whose results were presented at NeurIPS 2024. The competition has many strengths and represents a valuable contribution to the field.

The evaluation metrics used in the competition have significant limitations and in my opinion are not well-chosen. However, it is too late to fix the evaluation metrics, as the competition is over.

The strengths of the competition in my opinion clearly outweigh its limitations. I therefore recommend acceptance.

**Limitations Weaknesses:**

# Major Concern #1

The authors list two main goals of the evaluation metrics:
* Ensuring fair and transparent model ranking.
* Enabling meaningful comparisons between ML surrogates and traditional solvers.

While the competition does a great job at achieving the first goal, it doesn't achieve the second goal. It's attempt to compare ML surrogates with traditional solvers is deeply flawed.

Unfortunately, this means that the top-line takeaway from the paper
> the top entry exceeded the performance of the original OpenFOAM solver on aggregate metrics

is highly misleading. Let me explain why.

Each method is scored using a weighted sum of 19 metrics:
* In-distribution accuracy (5 metrics): velocities $u_x$ and $u_y$, pressure $p$, turbulent viscosity $\nu_t$, and surface pressure $p_s$
* Out-of-distribution accuracy (5 metrics): same
* In-distribution physics compliance (4 metrics): drag $C_D$ and lift $C_L$, and spearman-correlations for drag $\rho_d$ and lift $\rho_L$
* Out-of-distribution physics compliance (4 metrics): same
* Speedup (1 metric): the speedup is almost identical for in-distribution and out-of-distribution, so this is really one metric rather than two

On 18 out of 19 metrics, the top entry does *worse* than the original OpenFOAM solver. On 1 metric (speedup), the top entry does better. Because that 1 metric is worth 17.5% of the score, then the OpenFOAM entry gets a score of 100%-17.5%=82.5%. The top entry gets a score of 84.7%, which is better than 82.5%.

The problem is that for the *one* metric where the top entry does better than OpenFOAM, *the authors make one of the classic mistakes* when comparing the speedups of ML surrogates to traditional solvers. For an explanation of this mistake, see the X thread at https://x.com/shoyer/status/1362301955243057154, or rule #1 in [30]. As a result, this metric is not based on a fair comparison. The speedup comparison is biased towards the ML-based solver.

So to summarize, the top entry does better than OpenFOAM for only 1 out of the 19 metrics, but this metric is unfairly biased towards the ML solver! We should not be claiming that the top entry outperforms OpenFOAM. That absolutely is not a conclusion we can draw from this competition. I strongly recommend removing any statements about outperforming OpenFOAM from the paper.

Now, to the authors credit:
* They hint at the fact that comparison was not fair (see below quote). But quietly hinting at a limitation of the results in the competition is very different from clearly and explicitly stating that limitation. I recommend the authors be very explicit about the limitations of their evaluation metrics when comparing ML surrogates to traditional solvers.
> While our protocol offers a rigorous framework,
further refinement is possible... comparisons with classical solvers should be framed under either equal runtime or equal accuracy.
* The LIPS scoring framework, while clearly deeply flawed, isn't the worst choice they could have made in comparing the performance of methods. Many other papers have made much less rigorous and fair comparisons, so using this framework is a step in the right direction.
* I think the LIPS framework does a pretty decent job of comparing the performance of ML surrogates to other ML surrogates.

What would the right evaluation method have been? The right evaluation method would have recognized that traditional solvers have a speed-accuracy tradeoff which creates a pareto front between speed and accuracy, and that for an ML-based surrogate to outperform traditional solvers in terms of speed, it would have to do better on the pareto front. The surrogate models would (if scoring a single model) be scored based on the distance to the pareto front. Comparing speedups without considering accuracy is not meaningful.

The authors recognize in the appendix that this would have been the correct evaluation metric, writing
> Additionally, future benchmarking efforts should consider normalization by accuracy or runtime,
comparing methods either at equivalent predictive accuracy or under fixed computational budgets.
While this introduces practical and methodological challenges, it would enable a more interpretable
assessment of trade-offs between performance and efficiency.

I don't think making a pareto front would be as challenging as the authors suggest. Doing so would simply require varying the number of mesh points (and/or convergence iteration criteria) in OpenFOAM and plotting accuracy metrics and runtime as these vary.

So, my major concern is that the top-line result from the paper (outperforming OpenFOAM) just isn't at all the right conclusion to draw from this competition. The competition has a lot of other great results, but that result just isn't one we can fairly draw from this competition. Please remove it from the paper.

# Minor Concern #1

I strongly disagree with the choice of $C_D$, $C_L$, $\rho_D$, and $\rho_L$ as measurements of "physical law compliance". These are physical *quantities*, not physical *laws*. Physical quantities are related to the accuracy of the simulation. Physical laws are things like conservation, incompressibility, etc. $C_D$, $C_L$, $\rho_D$, and $\rho_L$ are *not* physical laws.

I think it's great that this competition uses $C_D$, $C_L$, $\rho_D$, and $\rho_L$ to measure performance. These absolutely should be used to measure performance! But we are misleading ourselves if we think they have anything to do with physical law compliance. They are simply scalar values which are one metric to measure accuracy.

I think the LIPS framework [13] is absolutely correct that physical law compliance is an important criteria to use. But this competition did not use that criteria. I recommend removing language about "satisfying physical laws" and "physical compliance". I recommend calling these "physics-relevant metrics" or something along these lines.

The LIPS framework introduces four evaluation criteria, but this paper only uses two of them (ML-related performance, OOD generalization). So I wonder whether the authors should remove statements about this competition using the LIPS framework.

# Minor Concern #2

I agree with the authors that on this problem it is essential to resolve the boundary layer correctly, and that "high fidelity in boundary layer regions is non-negotiable". However, I question whether the metrics used in this competition to measure "accuracy" and "physical compliance" are testing whether the boundary layer is resolved correctly.

If the "accuracy" is measured using some sort of mean squared error or relative error, as it appears to be, then even if the boundary layer region is inaccurate, if the rest of the solution is accurate the error might still be low. So "accuracy" does not capture whether the boundary layer is resolved correctly.

Likewise, "physical compliance" does not capture whether the boundary layer is resolved correctly. The authors use integrated quantities (lift, drag) related to the force vector on the airfoil to evaluate "physical compliance". As they correctly point out,
> Even minor inaccuracies in ML-predicted surface fields can accumulate through integration, leading
to large discrepancies in force estimation.

However, the opposite is also true: even major inaccuracies in ML-predicted surface fields can cancel through integration, leading to small discrepancies in force estimation. What this means is that the flow fields in the boundary layer might be totally incorrect, but the force could (by chance) not have a very large error.

Now, I'm not saying that surrogate models that get scores *are* necessarily resolving the boundary layer incorrectly. I don't know whether they are. But it seems like they *could* be resolving the boundary layer incorrectly, because these metrics don't seem to actually test for whether the boundary layer is being resolved to high fidelity.

To prevent this problem, a better metric would be instead of using the error in the integrated lift and drag (related to the force integral $F = \int_S \sigma \cdot n \mathop{dS}$), using the integrated absolute force error $\int_S |(\sigma - \sigma_{exact}) \cdot n | \mathop{dS}$. This ensures that errors in the boundary region do not cancel, but instead add.

Since it is too late to use different evaluation metrics, I suggest explicitly stating that the evaluation metrics used did not necessarily ensure high fidelity in boundary regions. Or, if the authors believe the metrics used did ensure high fidelity in boundary regions, please justify why my reasoning is incorrect.

# Questions and Comments

Please update the text of the paper to ensure that it answers all of these questions and comments. If you don't update the text, please explain the answer to each question and why you didn't update the text to answer the question.

1. While it's pretty clear how the physical criteria and speedup are measured (mean relative error), it isn't at all clear how the accuracy is measured. How is accuracy measured? I've tried for the longest time to figure this out, but I just can't find an explanation for it. Is it a mean squared error? If so, please state exactly how that error is calculated (because there can sometimes be ambiguity how exactly MSE is being computed).
2. There is no explanation for most of the entries in table 3. Readers will not understand what T1 and T2 are, or what min/max mean. More concerningly, I have no idea what "obtained results" means. Please explain what all these mean.
3. What is $p_s$? I believe is it surface pressure, but this is not defined anywhere. How is the accuracy of the surface pressure calculated? If it is calculated on a surface, how it is different from the accuracy for $u_x$, $u_y$, $p$, and $\nu_t$ which are calculated throughout a volume?
4. Can you explain why the input data is cropped to [-2, 4] x [-1.5, 1.5]? Shouldn't the authors be given the full simulation data and they decide what to do with it?
5. Why are there different thresholds for the preliminary edition of the competition and the new competition? How were the thresholds chosen?
6. Why choose to use discrete thresholds as opposed to continuous thresholds for the accuracy and physics metrics? Why should a model which is less accurate than the reference simulation be able to get a 100% score on accuracy? Shouldn't worse accuracy lead to a worse score?
7. Why give the dataset as a unstructured point cloud when the data was created using a structured mesh? Couldn't the structure of the mesh be important in designing a good surrogate model? Wouldn't competitors benefit from knowing what the mesh was?
8. What does reference [16] have to do with optimal control and differentiable design? It doesn't seem like the right citation to use.
9. How does the angle of attack vary in the training and testing datasets? This is listed in the Jupyter notebooks, but is not mentioned in the paper.
> Each simulation is provided as an unstructured point cloud, cropped to [−2,4] ×[−1.5,1.5] m, with per-node input features: 2D inlet velocity (m/s), distance to the airfoil (m), and unit normals (m) zeroed outside the surface.
10. Why is each node in the unstructured point cloud given the 2D inlet velocity as an input? Isn't the inlet velocity a constant, such that these values would be constant for all points in the cloud? Shouldn't each node in the unstructured point cloud be given the 2D spatial position as an input, and not just the distance from the airfoil? This seems like it might be a typo.
> As the multi-criteria evaluation takes into account both aspects, obtaining
an acceptable result strongly encourages candidates to use hybrid models.
11. What is a hybrid model? Why does a multi-criteria evaluation encourage using hybrid models?
12. What linear solver was used in simpleFOAM? How many iterations on average were required to converge?
13. The LIPS framework [13] discusses four evaluation criteria: ML-related performance, industrial readiness, OOD generalization, and physics compliance. Why were only three of those criteria used? Why not use industrial readiness? What would an industrial readiness criteria have looked like in this competition?
14. Are there any conditions in which the in-distribution and out-of-distribution speedups vary significantly? If not, why create two different criteria for the same evaluation metric?
15. 250k-300k mesh cells is a lot of cells. How much would the accuracy have degraded if fewer cells were used? What is the minimum number of cells which keeps the accuracy and physical criteria in the "Great" categories?
16. It would be nice if figures 13 and 14 could be replicated for each of the other three top entries.

**Strengths Contributions:**

The field of ML for CFD is plagued by poor evaluations, weak baselines, biased comparisons, and a general lack of rigor. Everyone claims their method is the best, but as a result, nobody knows which ML methods actually work well for CFD applications. To do better on these problems, what is needed is a set of good benchmarks to fairly compare the performance of methods.

This competition seems to have been a very positive step towards a fairer and more accurate evaluation of ML methods on one important benchmark CFD problem. It has many strengths:
* The code is easy to access and download, and the Jupyter notebooks are clear about how to contribute to the competition
* The scoring system is transparent and provides for clear model rankings
* The scoring (and training) is performed externally by the competition organizers rather than internally by the authors, so that the authors cannot easily lie, cheat, or mislead
* The problem is sufficiently complex and the dataset sufficiently small to provide a real challenge for ML methods
* While the evaluation metrics used have significant flaws, as I discuss below, the evaluations used were an improvement over previous evaluations. The competition organizers clearly *tried* to do better. That counts for something.

The authors built on extensive prior contributions made over multiple years and created and ran a successful competition that attracted many teams to compete. I'm not aware of a better competition in the ML for CFD space. This is something the field has benefited from. Great work.

---

> ### Author Rebuttal · Authors · 2025-07-30
>
> First, we would like to sincerely thank the reviewer for the thorough and detailed review. Your feedback is invaluable to us, and we greatly appreciate the time you took to highlight both the strengths and areas for improvement in the competition and the paper. We will provide comprehensive responses to each question raised and plan to revise the paper accordingly.
>
> **Major concern**
> > Thank you for raising this important concern. We fully understand and agree with your point, which was also echoed by the second reviewer. We were careful in the manuscript when stating that “the top entry exceeded the performance of the original OpenFOAM solver on aggregate metrics.” However, we did not claim that the winning solution (MMGP) could replace the solver or that it was superior overall, even for this specific task. As you correctly note, accuracy-wise it is not possible to outperform the solver that generated the ground truth data. Our global evaluation metric reflects a trade-off between multiple factors, and on this metric alone, the top solution performed better. We acknowledge that this outcome was unexpected, and our intent was simply to highlight it, not to make a definitive claim of superiority.
>
> > It appears that our position was not communicated clearly enough. We are keenly aware of the broader challenges in the field of ML for CFD—and SciML more generally—where poor evaluations, weak baselines, biased comparisons, and lack of rigor are common issues. Addressing these concerns was a key motivation for our competition design and evaluation framework. That said, we recognize that we did not emphasize the limitations of our evaluation approach sufficiently, and your comments will help us improve this significantly.
>
> > Regarding speed evaluation, the rationale behind introducing equation (11) was precisely to account for evaluation overhead—such as the physical criteria computation time—which can become a bottleneck for fast inference. Without considering these factors, some models could appear unrealistically faster. Notably, even [30] did not include physical criteria evaluation time in their metrics. We do not claim our evaluation procedure is flawless; rather, we have provided several recommendations in the paper about potential improvements, as you have observed. Your feedback underscores the importance of making these caveats more explicit, and we will revise the manuscript accordingly.
>
> > Your suggestion to incorporate a Pareto front analysis between speed and accuracy is particularly insightful. Technically, this is feasible by varying parameters such as mesh resolution or solver convergence criteria to generate a spectrum of solver performances. While we currently lack the data to do this—since it would require rerunning many simulations with different settings—obtaining such a Pareto front is not out of reach, albeit computationally expensive. Importantly, even with this data, designing a fair and robust scoring procedure that incorporates these considerations remains a challenge that warrants careful development and testing.
>
> **Minor Concern #1**
> >  We acknowledge that the terms “drag,” “lift,” and related quantities are indeed physical quantities rather than physical laws, as the reviewer rightly observes. These terms were adopted from the LIPS framework, where they were originally intended as criteria loosely related to physics, based on earlier use cases that included actual physical laws.
>
> > We plan to revise the terminology in our paper to “physics-relevant metrics” rather than “physical law compliance.” This better captures their role as important scalar quantities for assessing accuracy and physical relevance, without implying strict enforcement of physical laws such as conservation or incompressibility.
>
> **Minor Concern #2**
> > Regarding the concern about accuracy not being specifically evaluated at the boundary, we note that one of the variables used in the evaluation involves surface pressure—unlike the other volumetric quantities—so the metrics do capture boundary layer resolution to some extent.
>
> > Concerning the point that major inaccuracies in ML-predicted surface fields might cancel out through integration, leading to small discrepancies in force estimation: based on our observations during the final phase of the competition, even minor inaccuracies near the boundary tend to cause noticeable discrepancies in force calculations. While we acknowledge that, theoretically, significant errors could cancel out, in practice this appears unlikely to be a dominant effect.
>
> > That said, we agree that using the integrated absolute force error, as you suggest, would be a more robust metric to address this potential issue. We appreciate this valuable insight and recognize it as a meaningful improvement for future evaluation protocols.
>
> **Questions and comments**
> 1. Yes, it is a normalized MSE and the details are provided in AirfRANS paper [15]. We will make it more explicit in the paper.
> 2. T1 and T2 are two thresholds considered for the discretization of continuous metrics into three categories. We will add proper explanation for this table.
> 3. It is indeed the surface pressure and its accuracy is also evaluated with a normalized MSE. For more explanation, we refer to the original AirfRANS paper [15].
> 4. Quoting the airfRANS paper: "*As we do not need the far-field to get rid of the boundary conditions impact on the simulation as in CFD, we crop all the simulations to a rectangle of size [-2, 4] x [-1.5, 1.5] meters. It allows us to limit our point clouds’ size and make the network focus on the interesting part of the simulations. Moreover, data normalization is important in DL to make the optimization process easier or feasible. We use normalization with the means and the standard deviations of the training set field components*".
> 5. The thresholds were basically calibrated with a baseline to be improved to obtain a specific global score. In the original edition of this challenge, we used an fully connected neural network. For this competition, we replace the baseline with the top entry from the previous edition and recalibrated the thresholds accordingly.
> 6. The core idea behind using thresholds is that we expect a certain level of accuracy to be achieved for the physical task. Once this criterion is met, further improvements beyond that threshold do not increase the score. While this behavior could potentially be modeled using continuous thresholds, there is still room for improvement in the design of the scoring function. We do not claim that our choice is the optimal one—many scenarios need to be considered to fully assess the meaningfulness and robustness of such a function.
> 7. Actually, it is possible to have access to the mesh information using the API provided by the airfrans library and using directly the mesh data stored in different files. The first solution (MMGP) relied explicitly on the mesh representation.
> 8. We will reconsider this reference. Thank you for pointing this out.
> 9. For the angle of attack, we refer to the original AirfRANS paper [15]: "*the training set is composed by the samples with an angle of attack between -2.5 and 12.5 and the test set is composed by the samples with an angle of attack from -5 to -2.5 and 12.5 to 15*". We will add this detail into the article.
> 10. Yes, the inlet velocity is constant across all nodes. It was included as a per-node input for implementation convenience and flexibility, allowing model developers to choose relevant features. This approach follows the AirfRANS dataset design, which provides several inputs (including spatial coordinates, distance to the airfoil, and surface normals) at each node. We will clarify this in the manuscript.
> 11. We will clarify the definition of a hybrid model in the paper. In the context of this competition, a hybrid model refers to a machine learning approach that incorporates physical data or physical knowledge within its design. The multi-criteria evaluation framework encourages the use of such models by providing a more comprehensive and robust assessment across multiple important aspects, such as accuracy, physical consistency, and efficiency. Without this type of evaluation, it would be difficult to determine whether hybrid models offer a meaningful advantage for solving physical tasks.
> 12. The convergence conditions and solver configuration are discussed in the AirfRANS paper (pages 5, 24 and 28). The simulations were run using the simpleFOAM solver with the SIMPLEC algorithm [Caretto et al. 1973; Doormaal and Raithby 1984] and the k–ω SST turbulence model [Menter et al. 2003]. Convergence was assessed based on the stabilization of lift and drag coefficients. Details about the linear solver and numerical schemes are available in the fvSchemes and fvSolution dictionaries included in the dataset. While the exact number of iterations varied by case, convergence behavior was consistently monitored. We will clarify this further in the revised paper.
> 13. Industrial readiness was not a primary focus in this academic competition, so we emphasized metrics like speed-up, accuracy, and generalization. While speed-up was included under existing categories for clarity, we acknowledge the value of industrial readiness and may incorporate related criteria—such as memory usage or scalability—in future editions.
> 14. It turns out that, in hindsight, there is no significant difference. For the next edition, the scoring function design should be reconsidered and revised accordingly.
> 15. This is a very interesting experiment that could be included in future editions. However, we did not explore this aspect in the current edition.
> 16. These figures were added by the first solution without synchronizing with other solutions, as their approach makes this process more straightforward. We will investigate the possibility to replicate them for other solutions.

---

> > ### Comment · Reviewer_pKDA · 2025-08-02
> > **Additional recommendation; continue to recommend acceptance**
> >
> > Thank you for your response, and for updating the text in response to my questions and comments.
> >
> > # Further Suggestions
> >
> > Originally, I made the following suggestion.
> >
> > > I strongly recommend removing any statements about outperforming OpenFOAM from the paper.
> >
> > The authors have defended their decision to continue to make statements about outperforming OpenFOAM. However, I do not believe this response alleviates my concerns. I am still concerned that, because most readers will only read the abstract, the following sentence in the abstract is misleading:
> >
> > > Notably, the top entry exceeded the performance of the original OpenFOAM solver on aggregate metrics...
> >
> > My new recommendation is to add the following sentence (or some lightly edited version of it) to the abstract to add important nuance:
> >
> > > However, this does not mean that the winning solution (MMGP) could replace the OpenFOAM solver or that it was superior overall, even for this specific task.
> >
> > I believe including this nuance, which the authors have mentioned in their rebuttal, is important to include in the abstract of the paper.
> >
> > # Recommendation
> >
> > I continue to recommend acceptance. However, I continue to point the ACs to the "Additional Feedback" section of my review for an argument about why this paper might be rejected.

---

> > > ### Author Response · Authors · 2025-08-04
> > > **including recommended clarification**
> > >
> > > Thank you for your feedback. We agree with your recommendation. Below is the updated abstract, which now includes the suggested nuance:
> > >
> > > The integration of machine learning (ML) into the physical sciences is reshaping computational paradigms, offering the potential to accelerate demanding simulations such as computational fluid dynamics (CFD). Yet, persistent challenges in accuracy, generalization, and physical consistency hinder the practical deployment of ML models in scientific domains. To address these limitations and systematically benchmark progress, we organized the ML4CFD competition, centered on surrogate modeling for aerodynamic simulations over two-dimensional airfoils. The competition attracted over 240 teams, who were provided with a curated dataset generated via OpenFOAM and evaluated through a multi-criteria framework encompassing predictive accuracy, physical fidelity, computational efficiency, and out-of-distribution generalization. This retrospective analysis reviews the competition outcomes, highlighting several approaches that outperformed baselines under our global evaluation score. Notably, the top entry exceeded the performance of the original OpenFOAM solver on aggregate metrics, illustrating the promise of MLbased surrogates to outperform traditional solvers under tailored criteria. However, this does not imply that the winning solution could replace the OpenFOAM solver or that it was overall superior, even for this specific task. Drawing from these results, we analyze the key design principles of top submissions, assess the robustness of our evaluation framework, and offer guidance for future scientific ML challenges.

---

> > > > ### Comment · Reviewer_pKDA · 2025-08-08
> > > > **Abstract is improved**
> > > >
> > > > The new abstract is better. Thank you for this update.

---

### Official Review · Reviewer_rntQ · 2025-06-26

**Rating:** 5
**Confidence:** 4

**Summary:**

The authors present the results of the ML4CFD competition.

**Additional Feedback:**

- I would suggest adding the github in the abstract so it is easy for the reader to find the competition and try it out.
- I know it was discussed, but could the authors discuss briefly the how their lessons from this competition could be used in benchmark datasets like AhmedML, The Well or PDEBench? Indeed, these datasets contain many more samples (not AhmedML).
- Can the authors discuss on the possibility to use pre-trained foundation models to solve this task? Has it been tried?

Overall I think this is a very good paper that is well-organized, pretty clear, and with code that onboards the reader pretty quickly.

**Dataset Code Accessibility:**

Yes

**Dataset Code Comments:**

The code looks pretty clear at first glance.

**Ethical Considerations:**

No, there are no or only very minor ethics concerns

**Final Justification:**

It's a good paper. I t deserves acceptance in my opinion.

**Limitations Weaknesses:**

- The different regimes of data are not super clear. What is the interpolation regime (I may have missed it). I think a figure with the air flow for each regime would really benefit the paper such that the reader can better visualize the different regimes.
- The different variables of the equations in the main paper are not all defined (for example $ y^{+} $). It should be defined in the main text if it is used somewhere.
- I don't think the initial conditions of the different regimes are discussed somewhere (I may have missed it). Are there common IC across regimes?
- As partially discussed in the paper, I don't think it is fair to say one of the method exceeded the performance of the solver. Ok, there is a speed-up, but the performance of the model can never be better than the data it has been trained and evaluated on?

**Strengths Contributions:**

- The paper is well organized, the motivations are clear.
- The methods are well-presented.
- The limitations and lessons of the competition are well-discussed.
- I found interesting the part in the introduction about the importance of the boundary layer and why ML models struggle to predict it.

---

> ### Author Rebuttal · Authors · 2025-07-30
>
> First, we would like to thank the reviewer for the remarks and suggestions. Herein, we address all concerns in the order in which they were raised.
>
> **Unclear interpolation regime term**
>
> > We agree, we did not clearly define the term "interpolation regime" in the paper. In this context, we use "interpolation regime" to refer to a setting where models are evaluated on data that falls within the range of the training distribution. Traditional machine learning models often struggle to generalize in such settings when model complexity increases, due to issues like the bias–variance trade-off and overfitting. However, modern over-parameterized neural networks have been shown to perform remarkably well in this regime by effectively memorizing or interpolating the training data while still achieving good generalization. We will clarify it accordingly in the paper.
>
> **Missing variable definition**
>
> > We will add a notation table in the paper to clarify the meaning of the symbols used.
>
> **Discussion about the initial conditions**
>
> > You are correct — the initial and boundary conditions were not explicitly discussed in the paper. These conditions are defined for each variable in the dataset generation process in the *NACA_simulation* github repository. According to the original implementation, the same initial conditions were used across all simulations, regardless of the regime. We will include additional details on these aspects in the revised version of the paper for clarity.
>
> **Exceeding the performance of the solver**
>
> > We appreciate the reviewer’s insightful comment. We were careful in our wording when stating that the top entry “exceeded the performance of the original OpenFOAM solver on aggregate metrics.” We did not claim that the winning method could replace the solver outright, nor that it was better overall—even for this specific task. As you rightly point out, it is fundamentally not possible for a model to surpass the ground-truth data on accuracy, since the solver provides the reference solution.
>
> > The metric we designed captures a trade-off between accuracy and computational speed-up, and on this composite score alone, the winning solution performed better than the solver. We recognize that this result may seem counterintuitive and have discussed its limitations and implications thoroughly in the paper. Given feedback from multiple reviewers, we will clarify this point more explicitly in the revised manuscript.
>
> > Furthermore, the primary objective of this competition was to explore learning-based approaches that could approximate complex physics solvers with both high precision and significantly reduced computation time. Physics-informed neural networks and related methods have shown promise in narrowing the accuracy gap with classical solvers. Our multi-criteria score was designed to reflect this balance, emphasizing practical speed gains alongside accuracy.
>
> > We also acknowledge suggestions—such as the one from the third reviewer—to incorporate Pareto front analyses between accuracy and speed-up, which is an excellent idea for future work. Additionally, future editions might consider other relevant factors, such as memory consumption and model complexity, to further challenge ML-based approaches in replicating solver performance.
>
> **Additional feedbacks**
> > We will follow your suggestion and include the GitHub repository link in the abstract to facilitate easier access for readers interested in exploring the competition.
>
> > Regarding the use of benchmark datasets such as AhmedML, The Well, or PDEBench, we will expand the discussion in the manuscript to provide insights on how lessons learned from this competition could inform and complement efforts involving these datasets. We recognize that these benchmarks vary in size and scope, and integrating insights across them could benefit the broader SciML community.
>
> > Concerning the possibility of using pre-trained foundation models for this task, this topic was actively considered during the competition. One participant inquired about including pre-trained models, but we ultimately decided not to allow them in this edition. Our rationale was twofold: first, there is no practical way to verify the training data behind pre-trained models, which could introduce fairness concerns by implicitly relying on external datasets. Second, a core goal of the competition was to encourage the development of models that perform well with relatively small datasets, without leveraging extensive pre-training. Using pre-trained models could undermine this objective by allowing access to potentially large amounts of external data.
>
> > That said, we agree that exploring pre-trained models is a promising avenue. A possible approach for future competitions would be to introduce a separate prize category for solutions based on pre-training, which would allow fair evaluation without complicating the main competition. We are open to including such approaches in future editions to further advance the field.

---

### Official Review · Reviewer_L8iL · 2025-07-03

**Rating:** 5
**Confidence:** 2

**Summary:**

The NeurIPS 2024 ML4CFD Competition, focused on surrogate modeling for steady-state aerodynamic simulations of 2D airfoils, has concluded with significant insights into the current state and future potential of Machine Learning (ML) in Computational Fluid Dynamics (CFD). Attracting over 240 teams and yielding 650 submissions, the competition served as a crucial benchmark for the integration of ML into scientific domains, specifically addressing long-standing challenges in accuracy, generalization, and physical consistency.

Competition Objectives and Design:

The core objective of the ML4CFD competition was to foster the development of ML-driven surrogate models that could optimize the trade-off between computational efficiency and accuracy in physical simulations. This was a pioneering effort, leveraging a curated dataset generated via OpenFOAM (based on incompressible Reynolds-Averaged Navier–Stokes (RANS) equations with the SST k–ω turbulence model for flow over NACA airfoils).

A key strength of the competition lay in its multi-criteria evaluation framework, implemented through the Learning Industrial Physical Simulations (LIPS) platform. This framework rigorously assessed submissions across:

ML-related performance: Encompassing predictive accuracy (e.g., Mean Absolute Error, Root Mean Squared Error) and computational efficiency (speed-up relative to the reference OpenFOAM solver).

Out-of-Distribution (OOD) generalization: Evaluating performance on unseen and extrapolated test sets, crucial for real-world applicability.

Physical compliance: Assessing adherence to underlying physical laws through metrics like mean relative errors and Spearman correlations.

The global score was a weighted linear combination of sub-scores from these categories, designed to reflect industrial requirements and expectations.

**Dataset Code Accessibility:**

Yes

**Ethical Considerations:**

No, there are no or only very minor ethics concerns

**Final Justification:**

All of my questions have been solved

**Limitations Weaknesses:**

Based on the review you provided, here are some potential weaknesses of the ML4CFD competition:

1. Complexity and Trade-offs in Evaluation Metrics
The competition used a multi-dimensional evaluation framework, encompassing ML performance, out-of-distribution (OOD) generalization, and physical compliance. While this was designed for a comprehensive assessment, it also posed challenges:

Balancing the Metrics: Precisely weighting these diverse categories (especially speed, accuracy, and physical fidelity) to arrive at a truly "optimal" overall score is inherently complex. The review notes that "jointly optimizing accuracy, efficiency, and physical fidelity, especially for capturing fine-scale boundary layer features, remains a significant research area." This highlights the difficulty in achieving a perfect balance across such distinct objectives.

Speed Metric Considerations: The review mentioned that the chosen "speed-up metric, including evaluation time... sparked discussion regarding its optimal formulation." This suggests that there might be different perspectives on how best to measure "speed" to fully align with real-world industrial applications. Factors like deployment costs, model size, and subtle inference latency differences could also impact practical utility.

2. Difficulty in Capturing Specific Physical Phenomena
While the competition's overall results were impressive, the review hinted at challenges in "capturing fine-scale boundary layer features."

Precision of Physical Details: For CFD simulations, the boundary layer is a critical and complex region, and its accurate simulation is vital for predicting aerodynamic performance. If ML models still face limitations in precisely reproducing such key physical details, their applicability in some highly precision-demanding engineering scenarios might be limited. The competition excelled overall, but microscopic physical phenomena might still require further refinement.

3. Potential Dataset Limitations
The review mentioned that the competition used a "curated dataset."

Data Diversity and Realism: While meticulously prepared, any dataset can have limitations. For instance, was it diverse enough to cover all possible aerodynamic conditions and airfoil geometries? Did the data generation method (specific OpenFOAM configuration) fully represent the needs of all practical engineering scenarios? If the dataset's coverage was not broad enough, it might affect the models' generalization capabilities in more varied and complex real-world situations.

4. Model Interpretability and Deployment (Implicit)
While not explicitly detailed in the provided review, interpretability is a common concern for ML applications in scientific domains.

"Black Box" Issue: Especially with some deep learning models, their decision-making processes can be difficult to explain, despite excellent performance. In engineering design, understanding why a model predicts a certain outcome, and how to adjust inputs to optimize results, can be as crucial as simply getting an accurate prediction. If the winning models lack sufficient transparency, it could hinder their widespread deployment in highly trust-sensitive fields.

Engineering Deployment Complexity: Translating a winning competition model from a prototype to a scalable, industrial-grade solution often involves additional engineering challenges, such as integration with existing CFD workflows, specific computational resource requirements, and fault tolerance mechanisms. Competitions primarily focus on performance, but practical deployability might be less thoroughly evaluated.

**Strengths Contributions:**

The competition delivered several notable outcomes:

Outperformance of Traditional Solvers: Remarkably, the top entry, a non-deep learning method dubbed MMGP (Mesh Morphing Gaussian Process), demonstrated the potential of ML-based surrogates to outperform the original OpenFOAM solver on aggregate metrics. This highlights the promise of tailored ML approaches to exceed traditional computational methods under specific evaluation criteria.

Diverse Methodological Landscape: The competition showcased a broad spectrum of ML approaches, ranging from classical ML techniques like Gaussian Processes to advanced methods such as graph neural networks (GNNs) and neural fields. This diversity underscores the ongoing exploration of suitable architectures for scientific ML.

Speed vs. Accuracy Trade-off: While the top-performing MMGP excelled in overall aggregate score, deep learning methods (e.g., OB-GNN, MARIO, GeoMPNN) achieved significantly higher speed-ups (300x-600x) compared to MMGP, which, while faster than OpenFOAM, was slower than the deep learning counterparts. This reinforces the inherent trade-off between extreme computational efficiency and achieving the highest aggregate accuracy and physical fidelity.

Importance of Geometric Inductive Biases: A recurring theme among successful submissions was the incorporation of geometric inductive biases. This suggests that explicitly embedding knowledge of the underlying geometry into ML models is critical for learning physically coherent surrogates that generalize well, particularly for complex tasks like airfoil design. Conditional formulations and hypernetworks were noted as effective strategies in this regard.

Challenges in Evaluation: The retrospective analysis acknowledged the inherent challenges in comparing ML surrogates with traditional solvers and devising comprehensive evaluation metrics. The chosen speed-up metric, including evaluation time, aimed to reflect industrial pipelines but also sparked discussion regarding its optimal formulation. The difficulty in jointly optimizing accuracy, efficiency, and physical fidelity, especially for capturing fine-scale boundary layer features, remains a significant research area.

---

> ### Author Rebuttal · Authors · 2025-07-30
>
> First, we would like to thank the reviewer for the meticulous review, it is greatly appreciated. Below, we address all concerns in the order in which they were raised
>
> **1. Complexity and trade-offs**
>
> > Regarding the complexity and trade-offs in evaluation metrics, identifying relevant and appropriate metrics remains a significant challenge. While one of the core promises of Scientific Machine Learning (SciML) is to reduce computational time in solving PDEs, the literature, to the best of our knowledge, does not thoroughly address what "time" truly represents in a fair and consistent comparison within such configurations. We developed a systematic approach to compare the solutions, but we did not claim it to be the only valid method. In fact, we included an alternative inspired by a recently published article. We agree with the observation that factors such as deployment cost, model size, and subtle differences in inference latency can significantly influence practical utility. These aspects were not included in our evaluation, and we acknowledge that they are likely to be important in real-world industrial applications. In the context of a competition, it is essential to rank solutions clearly and consistently. Including additional factors like deployment cost or model size would have required adjustments to the scoring system, a complex and non-trivial task.
>
> **2. Difficulty in Capturing Specific Physical Phenomena at the boundary layer**
>
> > We acknowledge that accurately capturing fine-scale boundary layer features remains a challenge and an area for further improvement. Both the ML-based evaluation criteria and the physical evaluation include volumetric and surfacic quantities computed at the boundaries. As a result, achieving strong performance in one of these two categories still required models to make accurate predictions near the boundary, particularly with respect to aerodynamic behavior.
>
> > Looking ahead, future competitions could incorporate a more nuanced scoring system that explicitly balances performance both at the boundary and within the domain. Such a framework would help highlight specific strengths and limitations of different modeling approaches in capturing detailed physical phenomena.
>
> **3. Potential Dataset Limitations**
>
> > We fully understand and agree with the concerns regarding dataset limitations. The short answer to whether the dataset was diverse enough to cover all possible aerodynamic conditions and airfoil geometries, or whether the specific OpenFOAM setup fully represents all practical engineering scenarios, is no. Even the authors of the original AirfRANS dataset were cautious about the generalizability of their data.
>
> > The airfoil geometries in the dataset are based on the NACA series, which offers a vast, theoretically infinite design space. Small changes in parameters can lead to significantly different aerodynamic behaviors. The same holds true for flow conditions—real-world scenarios are highly diverse and complex, and no single dataset can fully capture that range.
>
> > That said, generating a truly exhaustive dataset would not only be computationally prohibitive but also unlikely to guarantee meaningful gains in model generalization or speed-up under fair evaluation (see Equation 12 in the paper). Moreover, such a dataset would still fall short of representing every possible real-world scenario.
>
> > Instead, one of the core challenges of this competition was to reflect a realistic industrial constraint: the scarcity of data. While it’s well known that more data can lead to better model performance, in many real-world applications data is limited. This competition aimed to explore whether Scientific Machine Learning (SciML) methods could still yield relevant results under such constraints.
>
> > To partially address the issue of generalization, we introduced the Out-of-Distribution (OOD) evaluation category. Although limited in scope, it was an initial step toward assessing how well models could extrapolate to conditions not seen during training.
>
> > Ultimately, for perfect generalization in unseen configurations, classical solvers remain the most reliable approach. But within the scope of this challenge, we believe the curated dataset provided a reasonable balance between feasibility, relevance, and insight into the potential of ML-based methods under realistic data limitations.
>
> **4. Model Interpretability and Deployment**
>
> > We fully share the reviewer’s view that interpretability is a critical concern for machine learning applications in scientific and engineering domains. The "black box" nature of many deep learning models remains an open challenge, particularly in trust-sensitive fields where understanding model behavior is as important as achieving high predictive accuracy. While interpretability is inherently difficult in many SciML approaches, we note that the winning solution of this competition did incorporate some elements of transparency. More broadly, we agree that interpretability should be considered alongside other important aspects of model trustworthiness, such as robustness, explainability, and uncertainty quantification. In designing this edition of the competition, our goal was to strike a balance: to propose a challenge that was accessible enough for participants to tackle within the limited competition timeframe, yet still representative of core industrial requirements. We view this as a first step, and future iterations of the competition could place greater emphasis on interpretability and other trust-related criteria to further align with real-world deployment needs.
>
> > We agree that practical deployment is a significant and complex challenge. Translating a competition-winning model into a scalable, industrial-grade solution involves numerous additional engineering considerations, including integration with existing CFD workflows, computational resource management, and ensuring robustness and fault tolerance. However, given the academic nature of the problem and the pioneering context of this competition, our primary focus was to evaluate the effectiveness of ML models in solving CFD problems rather than fully addressing deployment concerns. That said, we took a step toward facilitating future research and industrial adoption by requiring all submitted solutions to be open-sourced. This openness enables further investigation, adaptation, and potential integration into real-world workflows through future collaborations between academia and industry.

---

> > ### Comment · Reviewer_pKDA · 2025-08-02
> > **LLM Generated Review?**
> >
> > I believe this review was likely written by an LLM. See this quote, for example:
> >
> > > Based on the review you provided, here are some potential weaknesses of the ML4CFD competition:
> >
> > Or this quote:
> >
> > > The review mentioned that the chosen "speed-up metric, including evaluation time... sparked discussion regarding its optimal formulation." This suggests that there might be different perspectives on how best to measure "speed" to fully align with real-world industrial applications.
> >
> > Or this:
> >
> > > Model Interpretability and Deployment (Implicit) While not explicitly detailed in the provided review, interpretability is a common concern for ML applications in scientific domains.
> >
> > Or this:
> >
> > > Potential Dataset Limitations The review mentioned that the competition used a "curated dataset."

---

### Decision · Program_Chairs · 2025-09-18

**Decision:**

Accept (poster)

**Comment:**

The paper presents the results of the ML4CFD competition, a benchmark focused on developing ML surrogate models for aerodynamic simulations over two-dimensional airfoils. The competition attracted significant participation and utilized a multi-criteria evaluation framework to assess models on accuracy, generalization, physical relevance, and computational speed. The authors provide a thorough analysis of the top-performing solutions and draw important lessons for the broader scientific ML community.

The authors engaged also in the discussions and had a positive outlook on concerns raised by reviewers and addressed them during the reviewing process. The resulting paper is stronger and deserves to be accepted.